# Antigenic strain diversity predicts different biogeographic patterns of maintenance and decline of antimalarial drug resistance

**Qixin He[1]\*†, John K Chaillet[1]†, Frédéric Labbé[2]‡**

[1]Department of Biological Sciences, Purdue University, West Lafayette, United States; [2]Department of Ecology and Evolution, University of Chicago, Chicago, United States

**\*For correspondence:** heqixin@purdue.edu

†These authors contributed equally to this work

**Present address:** ‡CIRAD, UMRPVBMT, LaRéunion, France

**Competing interest:** The authors declare that no competing interests exist.

**Abstract** The establishment and spread of antimalarial drug resistance vary drastically across different biogeographic regions. Though most infections occur in sub-Saharan Africa, resistant strains often emerge in low-transmission regions. Existing models on resistance evolution lack consensus on the relationship between transmission intensity and drug resistance, possibly due to overlooking the feedback between antigenic diversity, host immunity, and selection for resistance. To address this, we developed a novel compartmental model that tracks sensitive and resistant parasite strains, as well as the host dynamics of generalized and antigen-specific immunity. Our results show a negative correlation between parasite prevalence and resistance frequency, regardless of resistance cost or efficacy. Validation using chloroquine-resistant marker data supports this trend. Post discontinuation of drugs, resistance remains high in low-diversity, low-transmission regions, while it steadily decreases in high-diversity, high-transmission regions. Our study underscores the critical role of malaria strain diversity in the biogeographic patterns of resistance evolution.

## eLife assessment

The study is an **important** advancement to the consideration of antimalarial drug resistance: the authors make use of both modeling results and supporting empirical evidence to demonstrate the role of malaria strain diversity in explaining biogeographic patterns of drug resistance. The theoretical methods and the corresponding results are **compelling**, with the novel model presented moving beyond existing models to incorporate malaria strain diversity and antigen-specific immunity. This work is likely to be interesting to malaria researchers and others working with antigenically diverse infectious diseases.

## Introduction

Prolonged usage of antimicrobial drugs almost always results in the emergence and spread of resistant strains (***zur Wiesch et al., 2011***). The history of falciparum malaria chemotherapy over the last hundred years witnessed a succession of the spread of resistance to five classes of drugs region by region (***Blasco et al., 2017***). However, the patterns of drug resistance evolution, such as the speed of establishment and equilibrium frequencies, differ drastically across different biogeographic regions. Even though de novo-resistant alleles are constantly generated, widespread resistant strains can almost always be traced back to two unstable transmission regions, that is, Southeast Asia (especially the Greater Mekong Subregion) and South America (***Ecker et al., 2012***; ***Dondorp et al., 2009***; ***Noedl et al., 2008***). While the frequencies of resistant genotypes often sweep close to fixation in these

**eLife digest** Drug resistance among strains of the parasites that cause malaria is a growing problem for people relying on antimalarial drugs to protect them from the disease. This phenomenon is global yet exactly how resistance emerges, spreads and persists in a population often differs greatly between regions, which can complicate malaria control projects. For example, discontinuing the use of antimalarials can lead to the frequency of resistant strains declining in an area, such as Africa, but persisting at high levels in others, including Asia and South America.

Gaining resistance often leads to parasites becoming less transmissible than other strains. When antimalarials are not used, sensitive strains usually outcompete their resistant counterparts. However, prolonged use of antimalarial drugs tends to eliminate susceptible strains, allowing the previously outcompeted resistant strains to dominate. The local dynamics of antimalarial resistance are also shaped by multiple other factors such as transmission levels (how common the disease is in the region), the type of antimalarial measures used (such as drugs and mosquito nets), or previous immunity the population may have developed to specific strains. While many computational models have been developed to capture these dynamics, they usually fail to include strain diversity – a parameter reflecting the number of malaria strains the immune system is exposed to. This parameter is important as parasites need to escape both host immunity and drugs in order to be successful.

To address this gap, He, Chaillet, and Labbé created a computational model to investigate how strain diversity, transmission levels and other related factors influence antimalarial resistance. The model was used to explore how the frequency of resistant and susceptible strains changes over time once antimalarial drugs are rolled out and then halted. These analyses show that in areas with both low strain diversity and low transmission levels, susceptible parasites are more likely to be wiped out from the population, leading to a high frequency of resistant strains that persist after drugs are discontinued. However, in high diversity and high transmission regions, susceptible strains can remain in the population. Therefore, when drug treatments are stopped, resistance levels are more likely to drop due to these parasites outcompeting the drug-resistant ones.

Overall, this work demonstrates how modelling approaches that include strain diversity can help inform public health decisions aimed at reducing antimalarial resistance. In particular, they can provide important insights into the control strategies that are best suited for a specific region, suggesting that in low transmission areas intensive drug treatment may contribute to resistance. Instead, preventative strategies such as eliminating mosquitos and preventing bites with bed nets may prove more beneficial at reducing transmission rates in such areas.

regions under persistent drug usage (*Chaijaroenkul et al., 2011*; *Plummer et al., 2004*), their frequencies are more variable in endemic transmission regions such as sub-Saharan Africa (*Talisuna et al., 2002*). More interestingly, while in high-transmission regions a steady decrease of resistant genotypes often ensues from reducing the particular drug usage (*Narh et al., 2020*; *Hemming-Schroeder et al., 2018*), resistant genotypes are maintained at high frequency in low or unstable transmission regions even after the abandonment of the drug for several decades (*Lanteri et al., 2014*).

Plenty of mathematical models have been developed to explain some, but not all, of the empirical drug resistance patterns. Various relationships between transmission intensity and stable frequencies of resistance were discovered, each of which has some empirical support: (1) transmission intensity does not influence the fate of resistant genotypes (models: *Koella and Antia, 2003*; *Masserey et al., 2022*; empirical: *Diallo et al., 2007*; *Shah et al., 2011*; *Shah et al., 2015*); (2) resistance first increases in frequency and slowly decreases with increasing transmission rates (models: *Klein et al., 2008*; *Klein et al., 2012*); and (3) valley phenomenon: resistance can be fixed at both high and low end of transmission intensity (model: *Artzy-Randrup et al., 2010*; empirical: *Talisuna et al., 2002*). Other stochastic models predict that it is harder for resistance to spread in high-transmission regions, but patterns are not systematically inspected across the parameter ranges (model: *Whitlock et al., 2021*; model and examples in *Ariey and Robert, 2003*). Under non-equilibrium scenarios, that is, where insecticides or bednets temporarily reduced transmission, reductions in resistance frequency were also observed (*Alifrangis et al., 2003*; *Mharakurwa et al., 2004*; *Myers-Hansen et al., 2020*). Differences in these model predictions can be attributed to three types of model assumptions: (1) whether

and how population immunity is considered, (2) how the cost of resistance is modeled, and (3) whether and how multiplicity of infection (MOI) is included. Although the great advances in malaria agent-based models (ABMs) enabled the inclusion of more detailed biological processes (*Maire et al., 2006*; *Masserey et al., 2022*; *He et al., 2021*; *Labbé et al., 2023*), the complexity of ABMs limits a direct application to analytical investigation. It is, therefore, critical to formulate a generalizable mathematical model that captures the most important biological processes that directly impact the survival and transmission of the parasites.

While most models have explored factors such as drug usage (*Koella and Antia, 2003*; *Klein et al., 2012*), treatment rate (*Masserey et al., 2022*), vectorial capacity (*Artzy-Randrup et al., 2010*; *Bushman et al., 2018*), within-host competition (*Bushman et al., 2018*; *Hastings, 2006*), population immunity (*Klein et al., 2008*; *Artzy-Randrup et al., 2010*), and recombination (*Curtis and Otoo, 1986*; *Dye and Williams, 1997*; *Hastings, 1997*; *Hastings and D'Alessandro, 2000*), strain diversity of parasites has not been explicitly considered in mathematical models of drug resistance. Yet, orders of magnitude differentiate antigenic diversity of *Plasmodium falciparum* strains among biogeographic zones and drive key differences in epidemiological features (*Chen et al., 2011*; *Tonkin-Hill et al., 2018*). Hyper-diverse antigens of parasites in sub-Saharan Africa emerged from the long-term co-evolutionary arms race among hosts, vectors, and parasites (*Volkman et al., 2001*). In endemic regions of falciparum malaria, hosts do not develop sterile immunity and can constantly get reinfected with reduced symptoms (*Day and Marsh, 1991*). These asymptomatic carriers of the parasite still constitute part of the transmission and serve as a reservoir of strain diversity (*Tiedje et al., 2017*; *Bonnet et al., 2003*) despite the fact that parasite prevalence decreases with host age in endemic regions (*Aron, 1983*). This age–prevalence pattern was attributed to acquired immunity after repeated infections and represented as different generalized immunity classes in disease dynamics models (*Dietz et al., 1974*; *Molineaux and Gramiccia, 1980*; *Klein et al., 2008*). Later advances in molecular epidemiology indicate the importance of strain-specific immunity (*Kaufmann et al., 1999*).

During the asexual blood stage, intra-erythrocytic parasites express adhesin proteins at the red blood cell surface that help mediate binding to the epithelial layers of vasculature to avoid the clearance by spleen during circulation (*Bull et al., 1998*). One of the major surface proteins, *P. falciparum* erythrocyte membrane protein 1 (*Pf*EMP1), is encoded by *var* genes, a gene family of 60 different copies within a single parasite genome (*Rask et al., 2010*). Immune selection maintains the composition of *var* genes between different strains with minimal overlap (*He et al., 2018*). In high endemic regions, many antigenically distinct strains (or modules of strains) coexist in the transmission dynamics (*Pilosof et al., 2019*). Whether the hosts have seen the specific variants of the *var* genes largely determines the clearance rate of the parasites (*Barry et al., 2011*; *Djimdé et al., 2003*). Therefore, it is reasonable to suspect that variation in host-specific immunity, acquired from exposure to local antigenic diversity, plays a key role in local transmission dynamics as well as the fate of resistance. Thus, under the same vectorial capacity, different strain diversity results in significant changes in population-level immunity and transmission intensity, and the ensuing epidemiological patterns, such as MOI, age–prevalence curve, and the ratio of asymptomatic infections (*Tiedje et al., 2017*; *Ruybal-Pesántez et al., 2022*). These changes, in turn, alter the fate of resistance invasion. Therefore, in addition to generalized immunity represented in earlier studies, models need to formally incorporate specific immunity.

Another challenging aspect for earlier models is whether and how multiclonal infections (those with MOI > 1) are considered. Due to malaria's long duration of infection (*Collins and Jeffery, 1999*), it is common for the host to carry infections that are contracted from separate bites, referred to as superinfections. Meanwhile, hosts can also receive multiple genetically distinct strains from a single bite, especially in high-transmission endemic regions (*Nkhoma et al., 2018*; *Wong et al., 2017*; *Henden et al., 2018*). Susceptible-infected-recovered (SIR) models that only consider non-overlapping infections (*Koella and Antia, 2003*; *Klein et al., 2008*; *Artzy-Randrup et al., 2010*) cannot incorporate within-host dynamics of strains explicitly, which strongly impacts the fitness of resistant genotypes (*de Roode et al., 2004*; *Bushman et al., 2016*). Other superinfection models employ complex structures or specific assumptions that make it hard to link MOI with strain diversity or host immunity (*Koella and Antia, 2003*; *Klein et al., 2012*).

Here, we present a novel ordinary differential equations (ODE) model that investigates how strain diversity and transmission potential influence disease prevalence, hosts' strain-specific and generalized

immunity, and the resulting MOI distribution. In this model, strain-specific immunity toward diverse surface proteins determines the probability of new infections. In contrast, generalized immunity of the hosts determines the likelihood of clinical symptoms. Hosts are less likely to show symptoms with repeated infections but can still be reinfected by antigenically new strains and contribute to transmission. Our modeling strategy combines the advantages of both the traditional compartmental epidemiological models (i.e., tracking transmission dynamics and population immunity responses to different levels of transmission intensity) (*Koella and Antia, 2003*; *Klein et al., 2008*; *Artzy-Randrup et al., 2010*; *Klein et al., 2012*) and population genetics ones (i.e., tracking within-host dynamics with detailed consideration of fitness cost and competition among strains) (*Curtis and Otoo, 1986*; *Dye and Williams, 1997*; *Hastings, 2006*; *Hastings, 1997*; *Hastings et al., 2002*). With varying strain diversity, transmission potential, resistance cost, and symptomatic treatment rates, we explore the key questions outlined above: whether strain diversity modulates the equilibrium resistance frequency given different transmission intensities, as well as changes in this frequency after drug withdrawal, and whether the model explains the biogeographic patterns of drug resistance evolution. We found that due to the feedback between transmission and host immunity, high equilibrium prevalence can only be achieved in transmission regions with high strain diversity. We observed a negative correlation between parasite prevalence and resistance frequency, regardless of resistance cost or efficacy. Post drug discontinuation, resistant frequency is maintained much longer in low-diversity regions than in high-diversity regions. We then verified the main qualitative outcome from the model against the empirical biogeographic patterns of chloroquine resistance evolution.

## Results
### Model structure
In the compartmental ODE model, hosts' strain-specific immunity ($S$) regulates infectivity of parasite strains, while generalized immunity ($G$) determines symptomatic rate (*Figure 1*; see model details in 'Methods' and Appendix 1). Hosts are tracked in different classes of generalized immunity ($G$) and drug usage status (untreated, $U$; treated, $D$). Hosts move to a higher $G$ class if they have cleared enough infections and go back to a lower class if they lose generalized immunity (*Figure 1B*: $G_i \rightarrow G_j$, *Figure 1—figure supplement 1*). Lower $G$ classes correspond to more severe and apparent symptoms, which increase the likelihood of being treated by drugs ($U \rightarrow D$), as evidenced from most impacted countries where children are the main symptomatic hosts (*Tiedje et al., 2017*). The population sizes of resistant ($PR$) or sensitive (wild-type; $PW$) parasites are tracked separately in host compartments of different $G$ and drug status. Since hosts can harbor multiple parasite strains, parasites are assumed to be distributed independently and randomly among hosts within the same compartment (*Anderson and May, 1978*). Parasites can move between the compartments via the movement of hosts that harbor them or can be added to or subtracted from the compartments via new infections and parasite clearance, respectively. $PW$ can be cleared by host immunity and drug treatment, while $PR$ can only be cleared by host immunity. However, $PR$ has a cost, $s$, in transmissibility, and the cost is higher in mixed-genotype ($s_{mixed}$) infections than in single-genotype infections ($s_{single}$) following *Bushman et al., 2016*; *Harrington et al., 2009*; *Bushman et al., 2018*.

Instead of tracking antigenic diversity explicitly, we assume parasites have $n_{strains}$ with unique antigen compositions at the population level. We incorporate specific immunity by calculating the probability of seeing a new strain given a $G$ class upon being bitten by an infectious mosquito,

$$\eta_i = (1 - \frac{1}{n_{strains}})^{\nu_i} \tag{1}$$

where $\nu_i$ is the average number of cumulative infections received and cleared by a host in class $G_i$, and is updated at each time step as determined by the immune memory submodel (see Appendix 1).

### Impact of strain diversity and transmission potential on disease prevalence
To avoid assuming an arbitrary level of strain diversity given transmission rate, we explored the impacts of the number of strains and transmission potential on prevalence separately across the empirical range observed in the field using our compartmental ODE model (*Figure 2A*). Specifically,

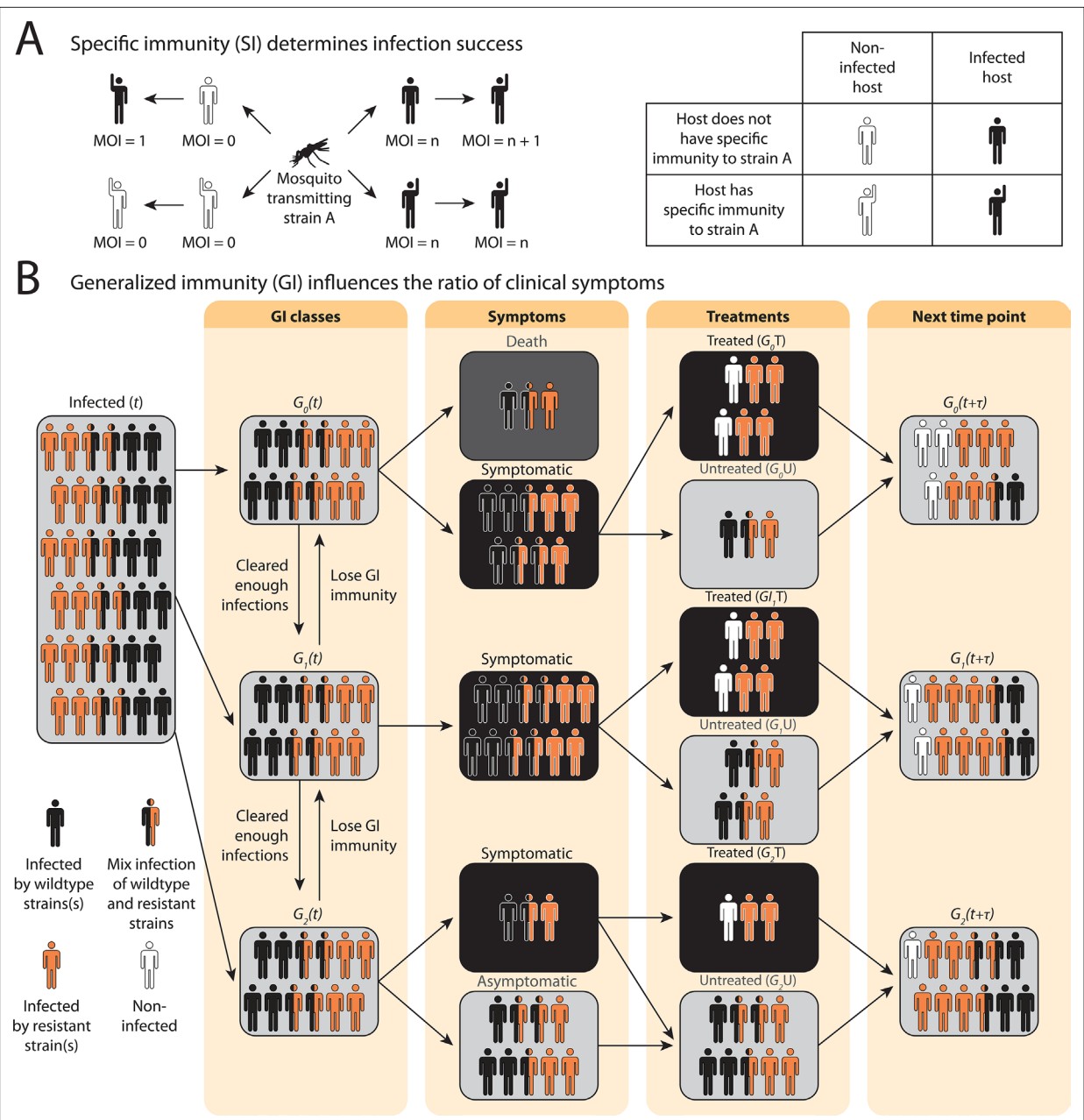

**Figure 1.** Schematic illustration of transmission rules and acquisition of host immunity within the compartmental ordinary differential equations (ODE) model (see *Figure 1—figure supplement 1* for a detailed representation of the compartment model). (**A**) Rules for new infections given the host's past infection history and current multiplicity of infection (i.e., multiplicity of infection [MO]). Upon transmission of a specific parasite strain A, if the host has had an infection of strain A in the past (hands raised), a new infection will not be added to the current MOI; instead, the infection will be considered cleared and added to the total number of cleared infections; if the host is new to strain A and does not have specific immunity to it (inferred from *Equation 1*), a new infection will be added (i.e., MOI increase by 1) as long as MOI does not exceed the carrying capacity of coexisting strains. (**B**) Rules of symptomatic infections and treatment in the different generalized immunity ($G$) classes. With increasing generalized immunity ($G$), hosts are less likely to show clinical symptoms. Hosts in $G_0$ have a risk of death in addition to symptomatic infections; Hosts in $G_1$ do not die from infections but show symptoms upon new infections; Hosts in $G_2$ carry asymptomatic infections most of the time with a slight chance of showing symptoms. Symptomatic infections result in a daily treatment rate that removes the infections caused by wild-type strains. Hosts that have cleared enough number of infections will move to the next $G$ class. Hosts will move back to a lower $G$ class when the generalized immunity memory is slowly lost if not boosted by constant infections.

The online version of this article includes the following figure supplement(s) for figure 1:

**Figure supplement 1.** Compartment model of drug resistance evolution.

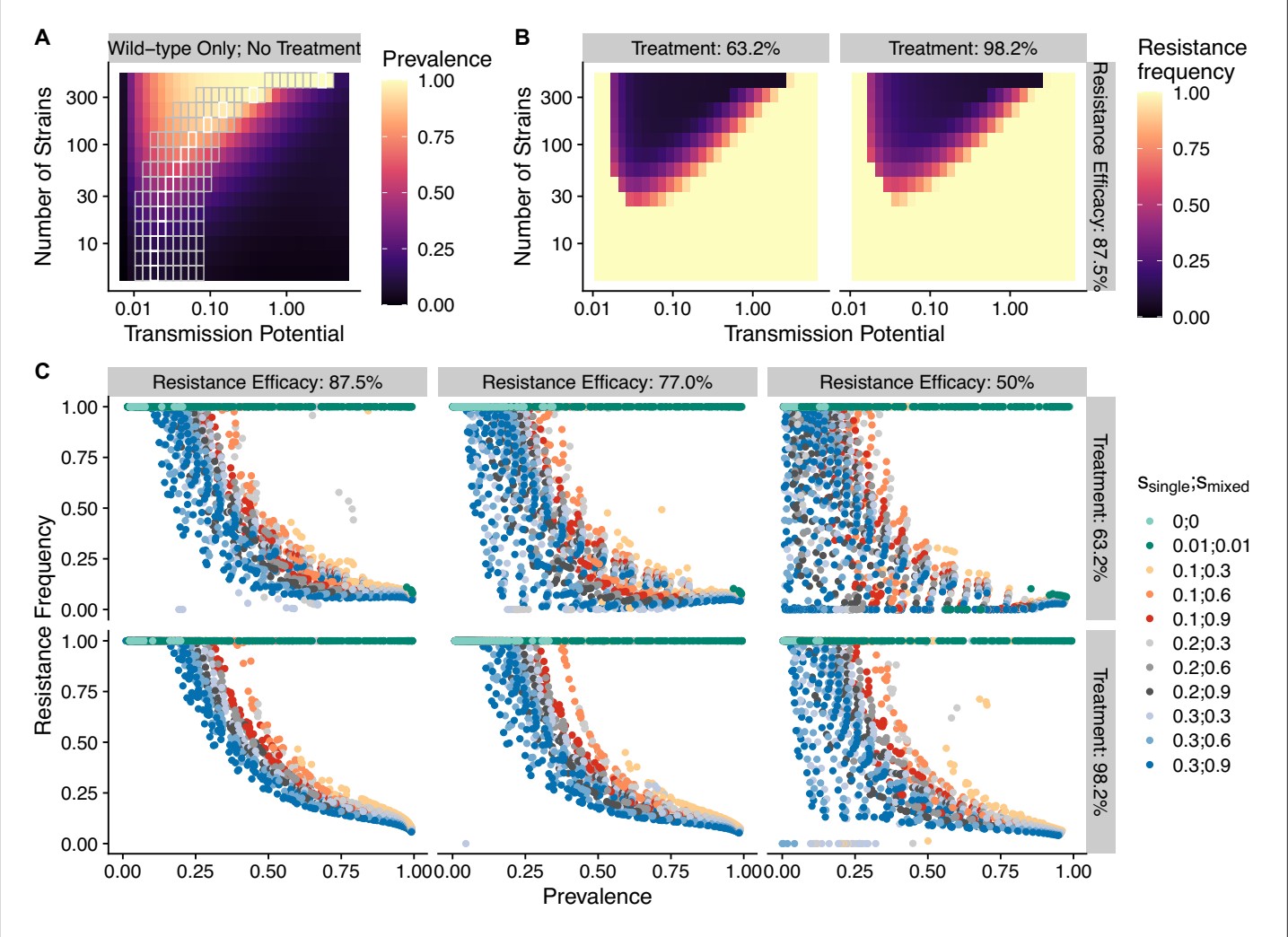

**Figure 2.** The frequency of resistance under varying strain diversity and transmission potential. (**A**) The heatmap shows a nonlinear parasite prevalence response given increasing transmission potential and the number of strains under no drug treatment, with warmer colors representing high prevalence and cooler colors representing low prevalence. X and Y axes correspond to increasing transmission potential and the number of strains in logarithmic scales. White tiles indicate the highest prevalence given a fixed number of strains. (**B**) The heatmaps show resistance frequencies under varying strain diversity and transmission potential at two levels of drug treatment rate, with warmer colors representing higher resistance frequency (in this example, $s_{single}$ = 0.1, $s_{mixed}$ = 0.9). A comparison between the prevalence pattern in (**A**) and resistance frequency in (**B**) reveals that high-prevalence regions usually correspond to low resistance frequency at the end of resistance invasion dynamics. (**C**) A negative relationship between parasite prevalence and resistance frequency. The color of the points indicates combinations of resistance fitness costs in hosts with resistant strains alone ($s_{single}$) or mixed infections of resistant and wild-type strains ($s_{mixed}$).

The online version of this article includes the following figure supplement(s) for figure 2:

**Figure supplement 1.** Prevalence given the combination of transmission potential and the number of strains from no treatment to high treatment rate for wild-type-only infections.

**Figure supplement 2.** Infectivity of a new infection as a function of the number of strains and mean immunity.

**Figure supplement 3.** Relationship between parasite prevalence and resistance frequency under full treatment (daily treatment rate $d1 = 0.2$).

**Figure supplement 4.** Relationship between parasite prevalence and resistance frequency under partial treatment (daily treatment rate $d1 = 0.05$).

the number of unique strains ranges from 6 to 447, which corresponds to a pool of 360 (typical of low-transmission regions) to 27,000 unique surface antigens (typical of sub-Saharan Africa) (**Chen et al., 2011**; **Tonkin-Hill et al., 2018**). Transmission potential refers to the product of vectorial capacity ($C$) and the maximum transmissibility between host and mosquito in one transmission cycle ($g$) (see 'Methods'). We model the pattern of transmissions through mosquito bites following a sinusoidal

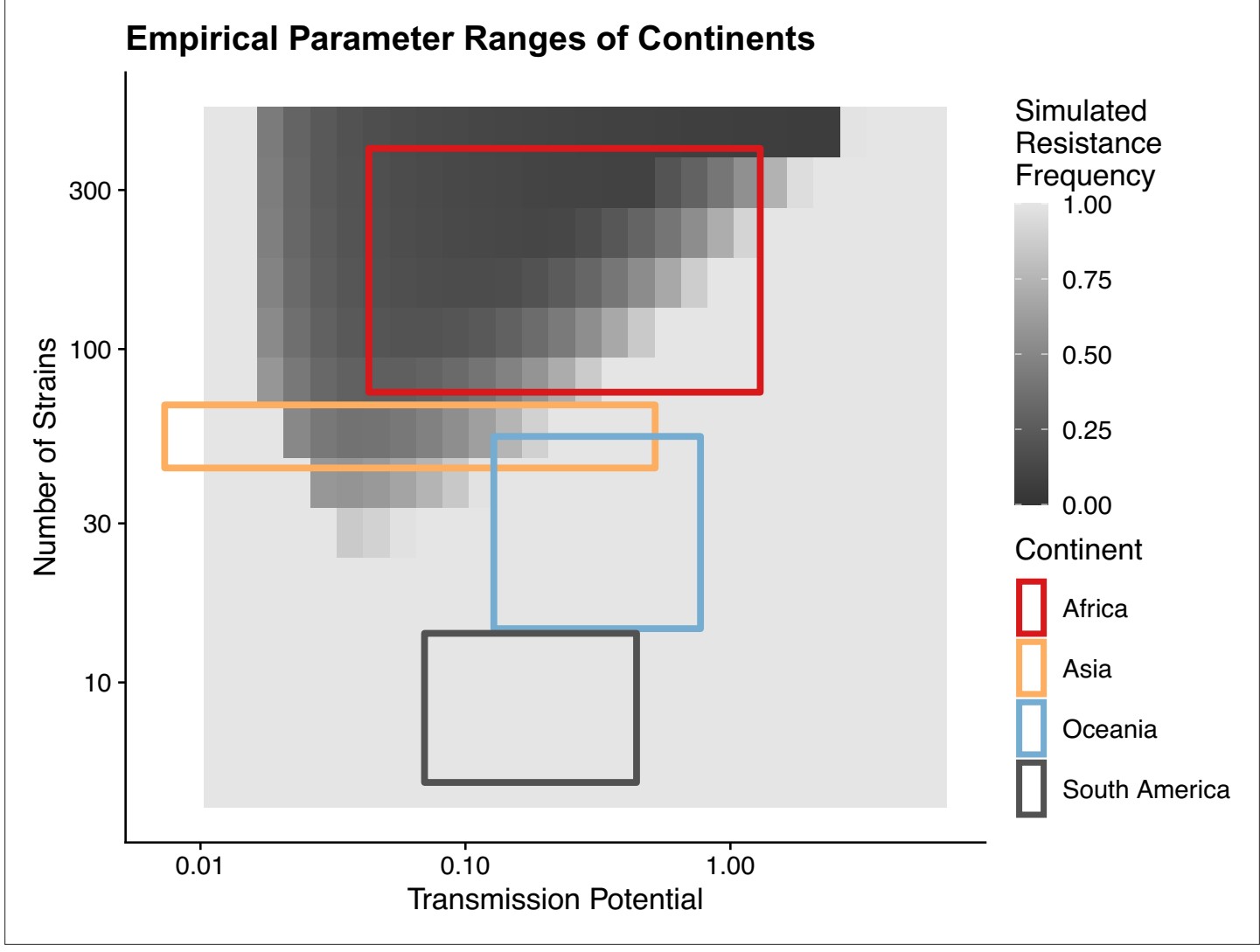

**Figure 3.** Empirical range of transmission potential and strain diversity. Squares denote the known minimum and maximum values of transmission potential and the number of strains from literature see *Tables 1* and *2* for parameter sources. We overlaid the empirical parameter ranges on the simulated equilibrium resistance frequency as a visual reference using the same parameters of *Figure 2B*. The empirical resistance frequency of these regions will depend on specific treatment rates and resistance costs, which is shown in *Figure 4*.

curve (Appendix 1), representing a peak transmission period in the wet season, and low transmission in the dry season annually, with a mean transmission potential from 0.007 to 5.8. Given $g$ of 0.08 (see 'Methods'), this range encompasses the lowest vectorial capacity to maintain a constant transmission to the level of high-transmission settings in Africa (*Garrett-Jones and Shidrawi, 1969*). We observe that the range of transmission potential that leads to the highest prevalence given a specific strain diversity increases from low diversity to high diversity (see gray area in *Figure 2A*, *Figure 2—figure supplement 1*). The prevalence decreases under drug treatment, but maintains the same relationship with strain diversity and transmission potential as that without treatment (*Figure 2—figure supplement 1*). This is consistent with the strain diversity being the outcome of long-term coevolution between parasite transmission and host immunity, whereby high-transmission regions usually correspond to high antigenic diversity and low-transmission regions exhibit low antigenic diversity (*Chen et al., 2011*; *Tonkin-Hill et al., 2018*). Therefore, for the following analyses, we focused on the parameter combinations within the gray area in *Figure 2A*, where diversity tracks transmission intensity. We then compared strain diversity and transmission potential pairing to the tentative empirical ranges of different continents (see 'Methods', *Figure 3*). As expected, strain diversity in Africa is much higher than in other continents, while transmission potential varies widely within continents, with overlaps

in medium ranges. Interestingly, while strain diversity in Africa and Asia tracks the range of transmission potential, Oceania and South America have lower strain diversity than expected by transmission potential.

## A negative relationship between disease prevalence and drug resistance frequency

To investigate resistanceinvasion, we introduce 10 resistant infections to the equilibrium states of drug treatment with wild-type-only infections and follow the ODE dynamics till the next equilibrium. In general, the frequency of resistance decreases with increasing parasite prevalence (*Figure 2B and C*), except for very low transmission potential, where resistance always fixes because wild-type strains cannot sustain transmissions under treatment (*Figure 2—figure supplement 1*, *Figure 2—figure supplements 3 and 4*). The fitness costs of single- and mixed-genotype infections, symptomatic treatment rate, and the efficacy of drug resistance only influence the slope of the relationship and the range of coexistence of resistant and wild-type parasites, but do not alter the negative relationship qualitatively (*Figure 2C*, *Figure 2—figure supplements 3 and 4*). Note that the negative relationship holds even when resistant genotypes have zero cost in transmissibility: they might still coexist instead of fix

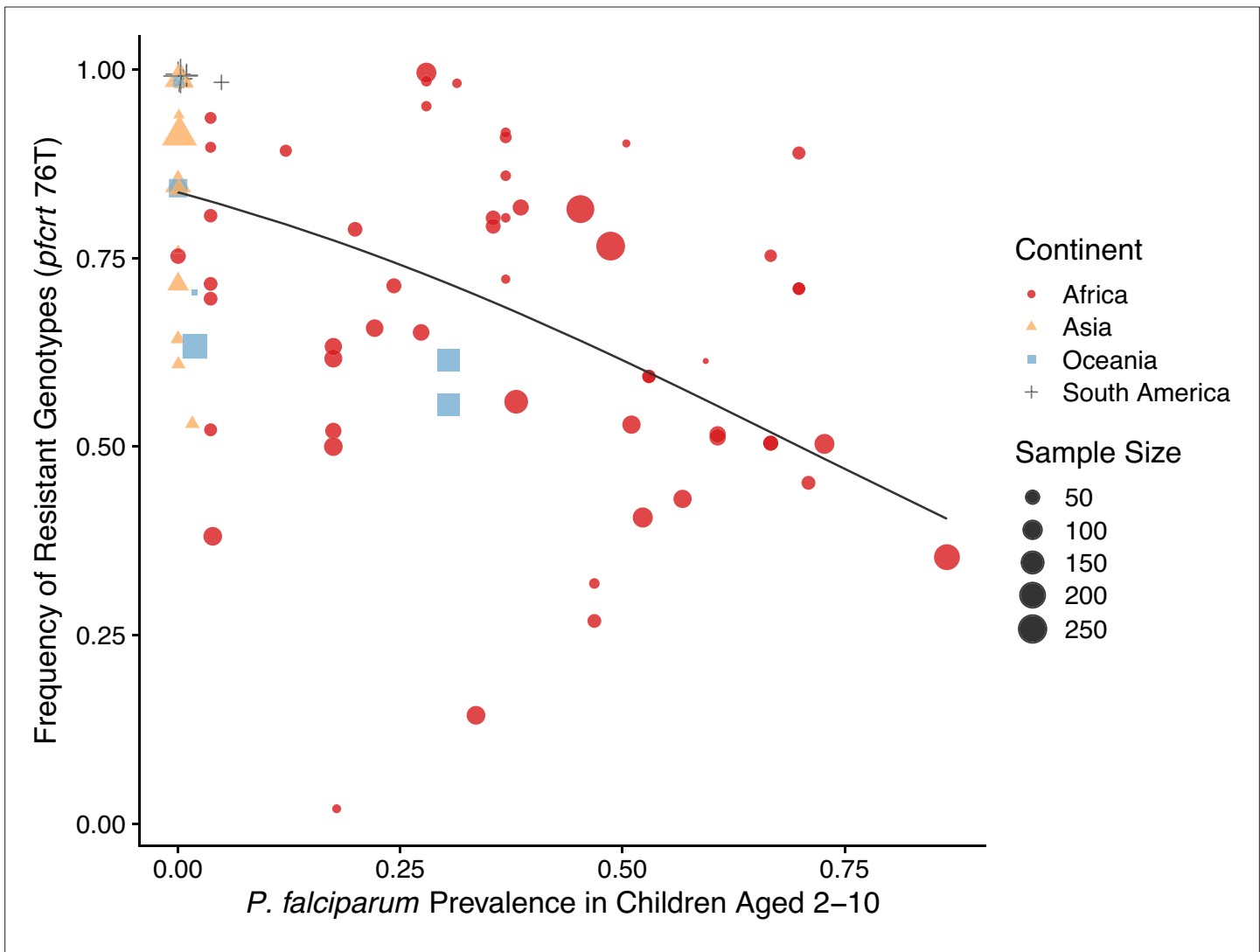

**Figure 4.** Global patterns of chloroquine-resistant genotype frequencies (*pfcrt* 76T) against *P. falciparum* prevalence in children between 2 and 10 years old. Sampling between 1990 and 2000 was included to ensure genotyping was performed largely before the policy switch of the first-line antimalarial drugs to ACT. Different shapes indicate samples from different continents, while shape sizes correspond to sample sizes for genotyping (see 'Methods' for details).

under very high disease prevalence (green dots in *Figure 2C*). Therefore, in the following sections, we only present results from one set of fitness cost combinations (i.e., $s_{single} = 0.1$ and $s_{mixed} = 0.9$ to be consistent with an earlier modeling study of parasite competition; *Bushman et al., 2018*).

The negative relationship between resistance and prevalence is corroborated by the empirical observation of the chloroquine-resistant genotype. The global trend of the critical chloroquine-resistant mutation *pfcrt* 76T follows an overall decline in frequency with increasing prevalence, which qualitatively agrees with the similar relationship from our model (*Figure 4*; beta regression, p value $< 2e - 16$). Samples from Asia and South America cluster around low-prevalence and high-resistance regions, with Asian samples having more variation in resistance, whilst samples from Oceania and Africa display a wide range of prevalence and resistance frequency. These characteristics could have emerged from our model dynamics given the parameter ranges of transmission potential and strain diversity of different continents (*Figure 3*).

## Dynamics of resistance invasion: Feedback between drug usage and host immunity modulated by strain diversity

The pattern of drug resistance and disease prevalence arises from the interaction between host immunity, drug treatment, and resistance invasion. In order to inspect the dynamics of resistance invasion in detail, we select a subset of strain diversity and transmission potential combinations as representative scenarios for the empirical gradient of low- to high-transmission settings for the following analyses (white squares in *Figure 2A*). So far, we have assumed that strain diversity and transmission potential may vary independently. However, in empirical settings, strain diversity is the outcome of long-term

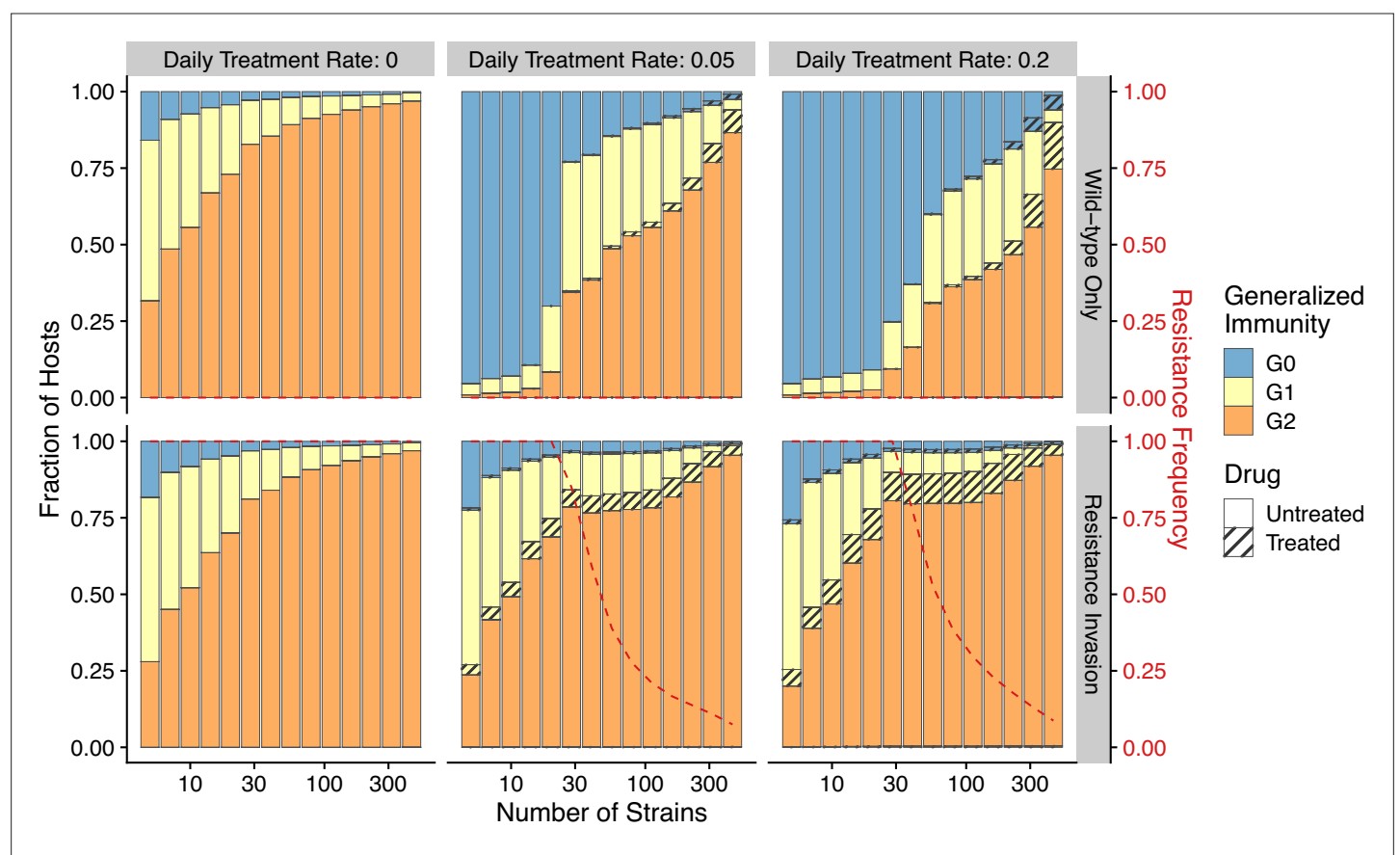

**Figure 5.** Relationship between host immunity, drug treatment, and resistance evolution. Fraction of hosts in different $G$ classes with increasing strain diversity and the corresponding transmission potential indicated by white circles in *Figure 1A* at equilibrium before drug treatment (left panel) or year 50 after the invasion of resistant genotypes (middle and right panels). Hosts under drug treatment are indicated by stripes. Red dotted lines show the corresponding frequency of resistance. The upper panel is generated under wild-type-only infections with increasing treatment rates. The lower panel represents resistance-only infections without treatment or resistant invasion under treatments.

coevolution between parasite transmission and host immunity, whereby high-transmission regions usually correspond to high antigenic diversity and low-transmission regions exhibit low antigenic diversity (*Chen et al., 2011*). Therefore, given the level of strain diversity, we picked the transmission potential that generates the highest prevalence. Under this constraint, the relationship between transmission potential and prevalence or diversity and prevalence is monotonic, in accordance with the prevailing expectation (*Figure 5A*). From low to high diversity/transmission, hosts' generalized immunity increases accordingly (higher fraction of hosts in $G_1$ or $G_2$ classes in *Figure 5*). When drug treatments are applied in a wild-type-only transmission setting, parasite prevalence is significantly reduced (*Figure 2—figure supplement 1*), as is host generalized immunity (*Figure 5A*, upper panel). A much larger proportion of hosts stay in $G_0$ and $G_1$ when effective drug treatment is applied compared to when there is no treatment. In addition, the proportion of hosts in drug-treated status increases under higher diversity. If instead the resistant genotype is present in the parasite population and starts invading when the drug is applied, hosts' generalized immunity is comparable at equilibrium to that of the no-treatment scenario (*Figure 5*, lower panel). The drug-treated hosts in $G_0$ and $G_1$ are comparable from low to high transmission, while the frequency of resistance decreases with increasing diversity (*Figure 5*, lower panel).

Temporal trajectories of resistance invasion show that parasite population size surges as resistant parasites quickly multiply (*Figure 6*). In the meantime, resistance invasion boosts host immunity to a similar level before drug treatment (*Figure 6*, upper panel). The surge in host immunity, in turn, reduces the advantage of resistant parasites, leading to a quick drop in parasite prevalence. Under a low-diversity scenario, wild-type parasites quickly go extinct (*Figure 6A*). Under high diversity, however, a high proportion of hosts in the largely asymptomatic $G_2$ creates a niche for wild-type parasites because the higher transmissibility of wild-type parasites compensates for their high clearance rate under drug treatment (*Figure 6B*). To summarize, the coexistence between wild-type and resistant genotypes in high-diversity/transmission regions reflects an interplay between the self-limiting resistant invasion and higher transmissibility of wild-type parasites as resistant invasion elevates the overall host immunity and thus the presence of a large fraction of hosts carrying asymptomatic infections.

## Response to drug policy change differs among high- and low-diversity scenarios

In our model, low-diversity scenarios suffer the slowest decline in resistant genotypes after switching to different drugs. In contrast, resistance frequency plunges quickly in high-diversity regions when the drug policy changes (*Figure 7*, *Figure 7—figure supplement 1*). Two processes are responsible for the observed trend. First, resistant genotypes have a much higher fitness advantage in low-diversity regions even with reduced drug usage because infected hosts are still highly symptomatic; this trend holds even if diversity is decoupled with transmission potential: given the same transmission potential, high-diversity scenarios have a faster percentage of reduction in resistance (see *Figure 7—figure supplement 1*). Second, if low transmission potential is coupled with low diversity, the rate of change in parasite populations is slower due to longer generation intervals between transmission events. This pattern corroborates similar observations across different biogeographic areas: while the transition of the first-line drug to ACT in Africa, such as Ghana and Kenya, resulted in a fast reduction in resistant genotypes, the reduction was only minor in Oceania, and resistant genotypes are still maintained at almost fixation in Southeast Asia and South America despite the change in the first-line drugs occurring more than 30 y ago (*Figure 8*).

## Comparison to a generalized-immunity-only model

Previous results demonstrate how transmission and antigenic diversity influence host immunity and hence the infectivity and symptomatic ratio, which determine the invasion success and maintenance of resistant genotypes. In order to confirm whether antigenic diversity is required to generate these patterns, we investigated a generalized-immunity-only model, in which infectivity of a new infection per $G$ class is set at a fixed value (i.e., taken as the mean value per $G$ class from the full model across different scenarios; see 'Methods'). We observe a valley phenomenon (i.e., resistance frequency is both high at the two ends of prevalence; *Figure 9*), which is qualitatively similar to *Artzy-Randrup et al., 2010*. Similarly, following the switch of first-line drugs, the medium-transmission region has the fastest reduction in resistance frequency, followed by the high- and low-transmission regions. This pattern

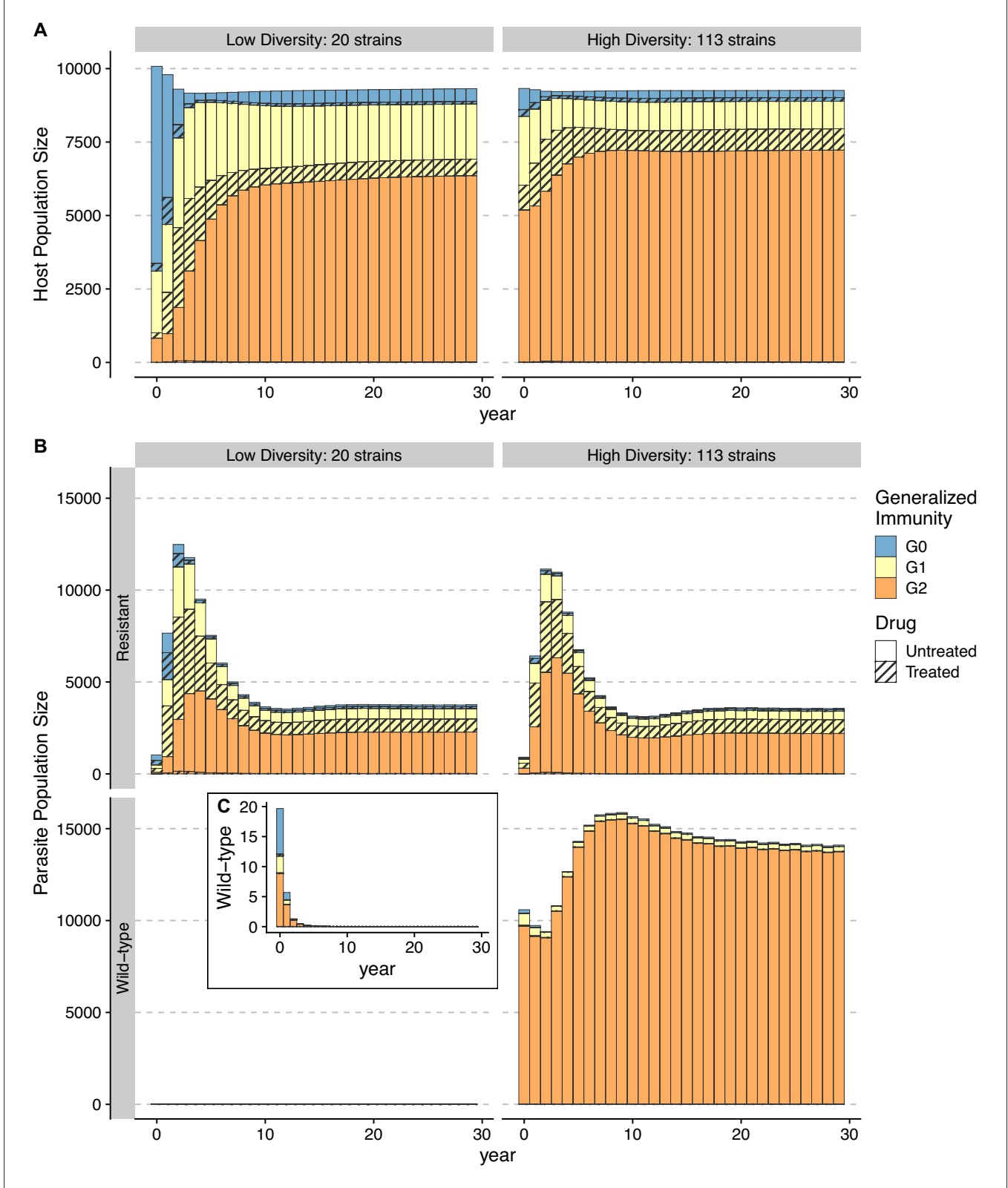

**Figure 6.** Temporal trajectories of resistance invasion. Host (**A**) and parasite dynamics (**B**) under resistance invasion are shown for lower ($n_{strains}$ = 20) and higher ($n_{strains}$ = 113) diversity under the same daily treatment rate of 0.05. Wild-type parasite population size is also presented in inset C with a smaller scale for clarity. Because drug treatment does not affect resistant parasites, they surge quickly after introduction, thus leading to more infections (upper panel of **B**). Hosts recovered from a large number of new infections move into higher $G$ classes (from year 1–8) (**B**). The higher specific immunity

*Figure 6 continued on next page*

reduces the infectivity of new strains, leading to a reduction of the resistant parasite population regardless of the diversity level (year 4–10; upper panel of **B**). Under low diversity, wild-type parasites quickly go to extinction C. Under high diversity, the less symptomatic $G_2$ class provides a niche for wild-type parasites to multiply (year 4–10), where the two genotypes coexist, with the wild-type parasite population size surpassing that of resistant ones. Meanwhile, resistant parasites dominate in hosts that are in $G_0$ and $G_1$ B.

also differs from that under the full model, where resistance in high-transmission regions reduces the fastest. When we compare how the host and parasite fraction in $G$ classes change with increasing transmission potential, we find that because the infectivity of bites does not decrease as transmission increases, the number of drug-treated hosts keeps increasing in the $G2$ class, resulting in the rising advantage of resistant genotypes (*Figure 9—figure supplement 2*). The comparison between the full model versus the generalized-immunity-only model emphasizes the importance of incorporating antigenic diversity to generate a negative relationship between resistance and prevalence.

## Discussion

In this article, we present a theoretical argument, built on the basis of a mechanistic model, as to why different biogeographic regions show variation in the invasion and maintenance of antimalarial drug resistance. While past models have examined the frequency of drug resistance as a consequence of transmission intensity and generalized immunity, these models, unlike ours, failed to reproduce the observed patterns of monotonic decreasing trend of resistance frequency with prevalence despite varying resistance costs, access to treatments, or resistance efficacy. This contrast stems from two main innovations of our model. First, its formulation directly links selection pressure from drug usage

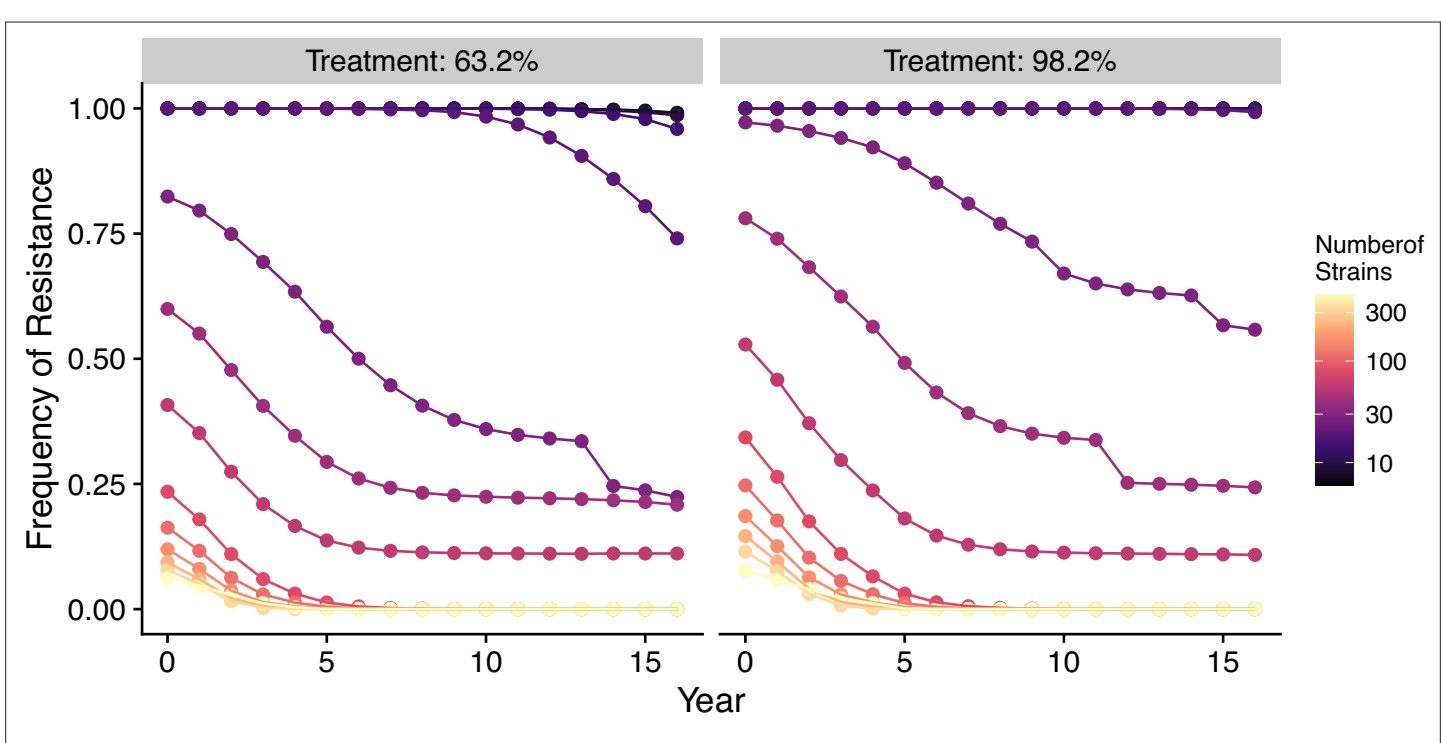

**Figure 7.** Changes in frequency of resistance after the first-line drug is changed. Each trajectory represents the mean resistance change from the combination of variables indicated by the gray area in *Figure 1A*. Color from cool to warm represents increasing diversity in strains. Here the usage of the drug, to which parasites have developed resistance, is reduced to 0.52, 0.52, 0.52, 0.52, 0.21, 0.21, 0.21, 0.21, 0, 0, 0, 0, 0, 0, 0 each year following the change in the treatment regime. The trajectory of reduction in resistant drug usage follows the usage survey in western Kenya from 2003 to 2018 (*Hemming-Schroeder et al., 2018*).

The online version of this article includes the following figure supplement(s) for figure 7:

**Figure supplement 1.** Percentage of reduction in resistance after 1 y of policy change in drug treatment as a function of transmission potential and the number of strains under different combinations of resistance costs ($s_{single}$; $s_{mixed}$).

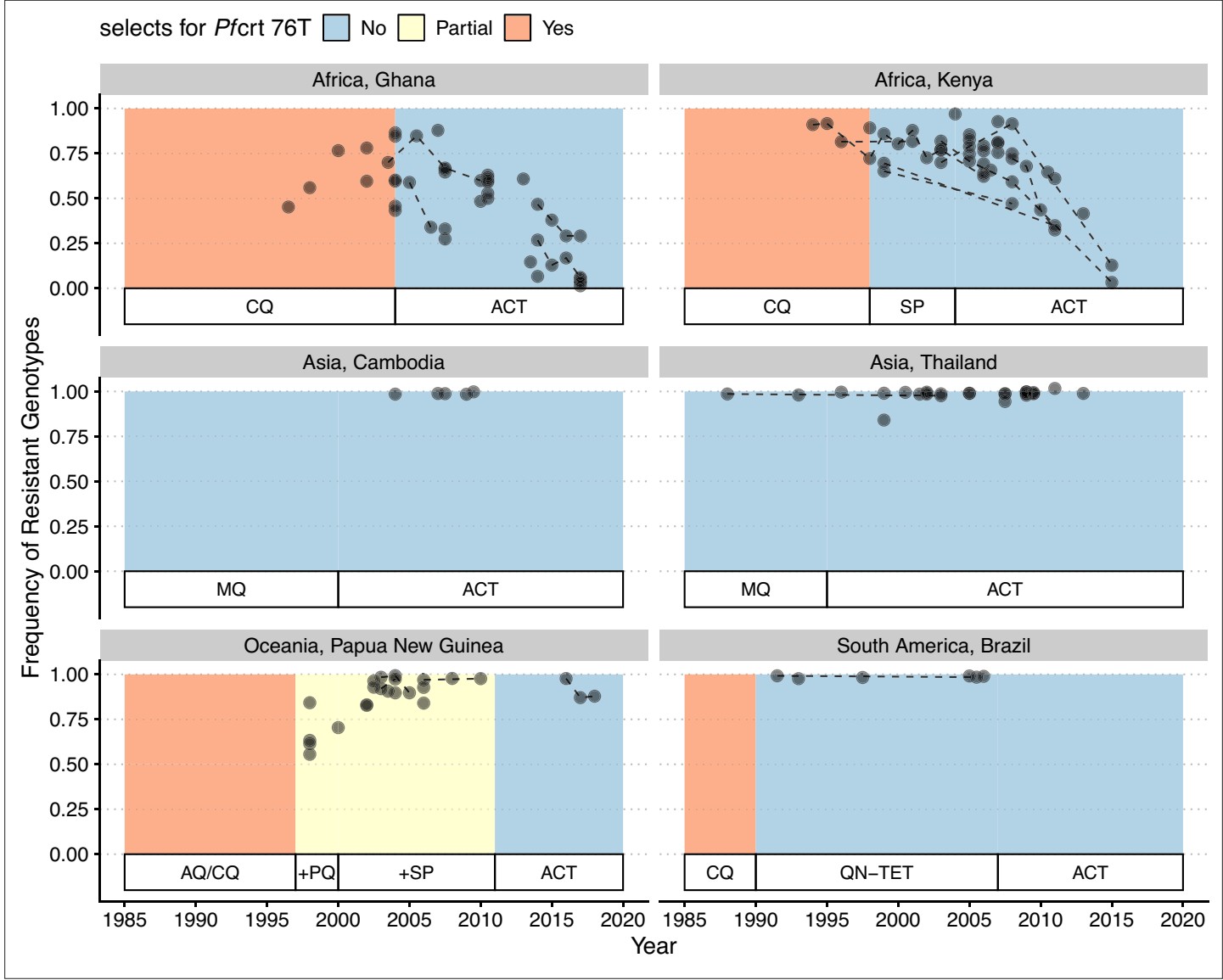

**Figure 8.** Changes in frequency of resistant genotypes across different biogeographic regions. Each circle represents one studied sample (at least 20 infected hosts) from one geographic location. Circles connected by dotted lines represent longitudinal samples from the same study. After the policy switch in first-line antimalarial drugs, frequencies of resistance decreased gradually in Africa, but maintained high in Asia, Oceania, and South America despite the policy change for more than 20 y. CQ: chloroquine; SP: sulfadoxine-pyrimethamine; MQ: mefloquine; AQ: amodiaquine; PQ: primaquine; QN-TET: quinine + tetracycline; ACT: artemisinin-based combination therapy.

with local transmission dynamics through the interaction between strain-specific immunity, generalized immunity, and host immune response. Second, this formulation relies on a macroparasitic modeling structure suitable for diseases with high variation in cooccurring infections and strain diversities (*Anderson and May, 1978*). Hosts are not tracked as infected or susceptible; rather, the distribution of infections in hosts of different immunity classes is followed so that within-host dynamics of parasites can be easily incorporated.

In essence, the dynamics of resistant genotypes of a single locus are governed by two opposing forces: the selective advantage from drug usage and the cost of resistance. Both forces emerge, however, from local transmission dynamics, contrary to many earlier population genetics or epidemiological models that set these as fixed parameters. For example, when a fixed fraction of hosts is assumed to be drug-treated upon infection (e.g., in *Curtis and Otoo, 1986*; *Dye and Williams, 1997*; *Hastings, 1997*; *Koella and Antia, 2003*), the frequency of resistance is found to be unrelated to transmission intensity or requires other mechanisms to explain why resistance is prevalent in

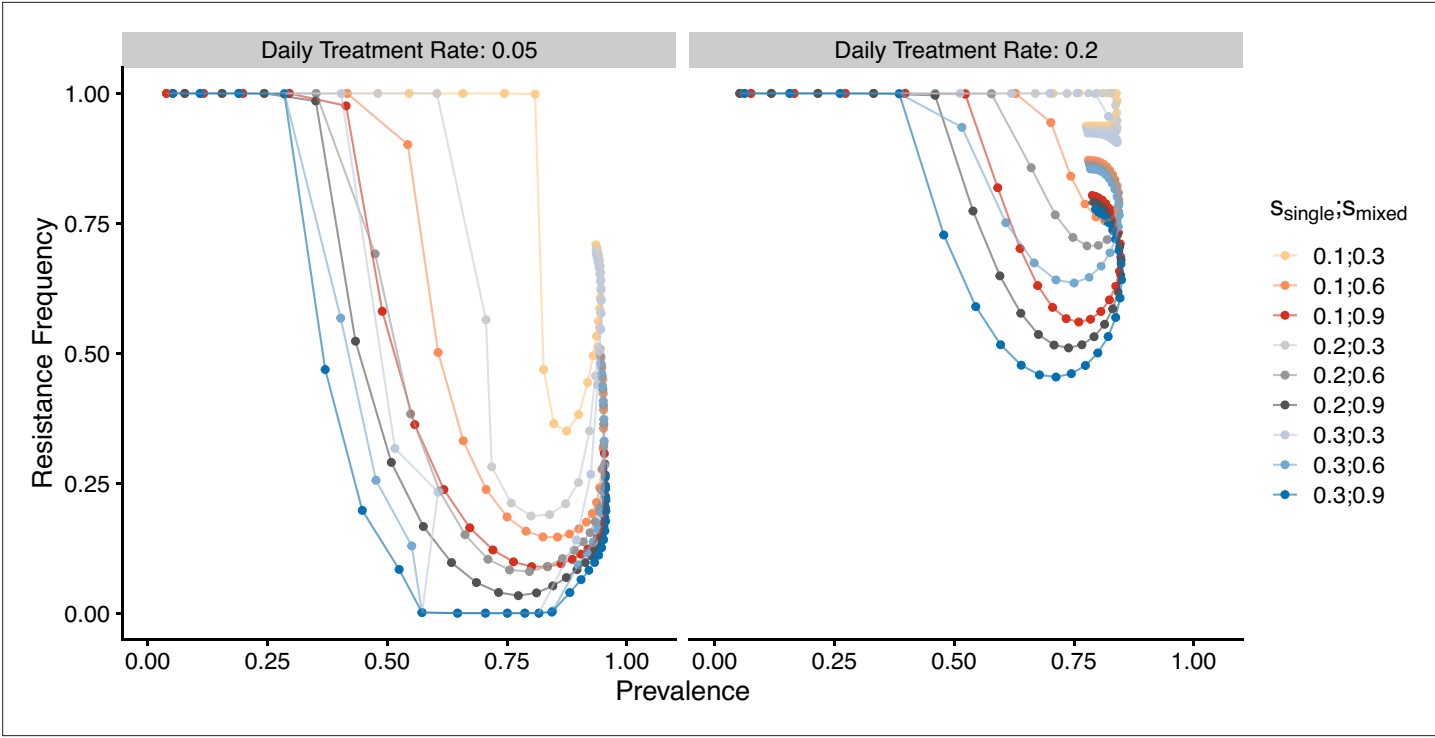

**Figure 9.** Relationship between parasite prevalence and resistance frequency for the generalized-immunity-only model. Paths are connected from low transmission potential to high-transmission potential. Colors represent different combinations of single-genotype infection cost and mixed-genotype infection cost of resistant parasites.

The online version of this article includes the following figure supplement(s) for figure 9:

**Figure supplement 1.** Changes in frequency of resistance after the first-line drug is changed in the generalized-immunity-only model.

**Figure supplement 2.** Relationship between host immunity, drug treatment, and resistance evolution for the generalized-immunity-only model.

low-transmission regions. Later models recognize the importance of clinical immunity gained through repeated reinfections (analogous to the $G2$ class in our model) in reducing drug usage (*Klein et al., 2008*; *Artzy-Randrup et al., 2010*). Countries with different access to treatment (i.e., different treatment rates of symptomatic patients) also influence the net advantage of resistance (*Masserey et al., 2022*). However, in these models, the infectivity of new bites constrained by antigen diversity is not considered such that under high transmission the clinically immune class still receives numerous new infections, and the lowered symptomatic rate does not offset the amount of drug treatment due to new infections, giving rise to the increasing resistance prevalence at the high end of transmission potential (see *Figure 9* and *Artzy-Randrup et al., 2010*). In contrast, in our model the selective pressure from drug treatment not only depends on the host ratio in the clinically immune class, but also on the infectivity of new bites regulated by specific immune memory. Therefore, when the host population suffers a high parasite prevalence, most hosts have experienced many infections and have entered the clinically immune class, where the drastically reduced infectivity coupled with the reduced symptomatic rate result in an overall reduced drug treatment per infection, mitigating the advantage of resistance.

Cost of resistance in terms of its form and strength is a complicated topic by itself. On the one hand, replication rates of resistant parasites are consistently found to be slower such that they produce less parasitemia during the infection than wild-type parasites (*Bushman et al., 2016*; *Koella, 1893*; *de Roode et al., 2005*). On the other hand, field studies also show that the transmissibility could be partially compensated by a higher gametocyte production (reviewed in *Koella, 1998*). Here we assume resistant parasites have lower transmissibility, but the cost differs between mixed- vs. single-genotype infections. Empirical and modeling studies *Bushman et al., 2016*; *Bushman et al., 2018*; *de Roode et al., 2004* have shown that within-host competition between resistant and wild-type infections results in a higher cost for resistant infections than in single-genotype infections. This

phenomenon could potentially prevent resistance establishment under high-transmission settings where mixed-genotype infections are more common (*Bushman et al., 2018*). However, we did not find that the higher cost in mixed-genotype infections influenced the qualitative pattern of a negative relationship between transmission intensity (represented by parasite prevalence) and resistance frequency. In addition, an equal cost in mixed- vs. single-genotype infections also produced a lower frequency of resistance at high transmission in the full model, but not in the GI-only model, indicating that within-host competition will exacerbate the disadvantage of resistant parasites under high transmission, but does not generate the negative correlation. The temporal dynamics of resistance invasion showed that the self-limiting property of resistant parasites creates a specific niche for wild-type infections to coexist. Specifically, as resistance invades, hosts experience more infections, leading to higher generalized immunity. Wild-type infections will then dominate in the lower symptomatic class because they have higher transmissibility.

The inclusion of strain diversity in the model provides a new mechanistic explanation as to why Southeast Asia has persisting resistance to certain antimalarial drugs, including chloroquine, despite a lower transmission intensity than Africa. In these regions with low strain diversity, parasites cannot repeatedly reinfect hosts. Therefore, clinically immune hosts do not carry infections very often. Thus, in our model resistant strains reach fixation or near-fixation regardless of the actual transmission potential, and upon removal of the drug pressure, these regions continue to maintain high levels of drug resistance for a prolonged time. In contrast, high-diversity regions (e.g., Africa) should show a wide range of resistance frequency depending on how antigenic diversity is matched with local vectorial capacities and should respond more rapidly to changing drug pressures. These results are partially corroborated by a comparison with regions that have higher transmission potential than Southeast Asia but low diversity (e.g., Papua New Guinea) (*Chen et al., 2011*; *Figure 3*). The resistance trends for Papua New Guinea behave most similarly to those for Southeast Asia, suggesting that strain diversity, instead of transmission potential, is key to predicting trends in drug resistance frequency. When diversity is less than expected by transmission potential, most mosquito bites have low infectivity, and most infections only occur in hosts with lower generalized immunity. Therefore, resistant genotypes will help ensure disease transmission in these symptomatic hosts and be strongly selected to be maintained.

As comprehensive as the model is, it still has some limitations. First, it currently assumes that a single locus determines resistance. If resistance is encoded or augmented by two or more loci (e.g., ACT or SP), past population genetic models demonstrate that rates of recombination could strongly influence the spread and maintenance of resistance (*Dye and Williams, 1997*; *Hastings, 2006*). Recent models have shown that preexisting partner-drug-resistant genotypes promote the establishment of Artemisinin resistance (*Watson et al., 2022*). However, as recombination is one of the potential reasons why multilocus resistance has delayed appearance in high-transmission regions, the incorporation of recombination is not expected to alter the negative relationship between resistance and prevalence. These earlier population genetics models of drug resistance posit that a high selfing rate in low transmission ensures high linkage among multilocus resistance, promoting their higher frequencies (*Dye and Williams, 1997*; *Hastings and D'Alessandro, 2000*; *Hastings and Donnelly, 2005*). Thus, adding multilocus resistance is expected to augment the negative correlation between resistance and prevalence. Expansion of the current model to include multilocus resistance will shed light on this prediction.

Second, our deterministic compartmental model does not consider several sources of variation in parasite transmission. Genetic drift is not incorporated in the model, which could influence the variation of strain frequencies at the population level due to severe bottlenecks during vector transmission (*Wong et al., 2018*). Demographic stochasticity would be more likely to impact low-transmission areas during the resistance invasion, while less impacting the biogeographic patterns for resistance maintenance. Our model also assumes that parasites are independently and randomly distributed in hosts, while the negative binomial distribution (NBD) is widely used in macroparasitic models (*Anderson and May, 1978*). Empirical evidence of parasite burdens is usually over-dispersed in that relatively few members of the host population harbor the majority of the parasite population (*Anderson and Gordon, 1982*; *Churcher et al., 2005*; *Grogan et al., 2016*). In our model, we argue that despite the MOI within each $G$ class being Poisson distributed, the population-level MOI distribution is over-dispersed as hosts in the $G2$ class are much less likely to be infected than in $G1$ or $G0$ (*Figure 2—figure*

*supplement 2*) and hosts in drug-treated classes have lower MOI than untreated classes as they harbor mostly resistant parasites only. By discretization of host classes and parasite types, we considered over-dispersion at the population level. Future models could expand on the NBD for individual classes by fitting empirical data from different age classes. The assumption of independent distribution of parasites also implies homogeneous within-host selection and equal frequency of the same genotype strains. In reality, within-host strain frequency will vary depending on the time of each infection, strain similarities, and host immunity to specific antigens. These processes will generally increase the variation of resistant genotype frequencies at the population level, but should not impact the overall biogeographic pattern inferred here.

Lastly, our model assumed a random association between resistant genotype and antigenic diversity. In reality, in the early stage of invasion, the resistant genotype should have a limited antigenic background until it becomes widespread. In an agent-based stochastic model, *Whitlock et al., 2021* found that selection for high antigenic variation in high transmission slows the spread of resistance. The interference of immune selection and resistance might serve as an additional reason why resistant parasites are at lower frequencies in high-transmission settings. Future stochastic models are desirable for quantifying the dynamics of interactions between antigenic variation and resistant loci under different epidemiological settings.

It is also to be noted that the trend found in our model predicts an equilibrium state of resistance frequency under persistent drug usage, which cannot be extrapolated to transient dynamics of new drug introduction. As shown in *Figure 7*, a fast sweeping phase is always associated with a new introduction of resistant genotypes in both low- and high-diversity regions. Therefore, we focused on empirical comparison to Pfcrt 76T because this mutation is essential for chloroquine resistance (*Ecker et al., 2012*) and chloroquine has been heavily used as first-line drugs for years in most countries.

In sum, we show that strain diversity and associated strain-specific host immunity, dynamically tracked through the macroparasitic structure, can predict the complex relationship between transmission intensity and drug resistance frequencies. Our model implies that control protocols should vary from region to region and that there is no one-size-fits-all cure for malaria control worldwide (*Rasmussen et al., 2022*). In regions of low prevalence, such as Southeast Asia, long-term goals for malaria prevention will likely not be aided by intensive drug treatment (*Delacollette et al., 2009*; *Imwong et al., 2020*). In these regions, elimination of falciparum malaria through vector control measures could proceed with little effect on drug resistance levels, whereas continual drug treatment will almost certainly cause fixation or near-fixation of resistance for a prolonged period of time, even after discontinuation of one drug. In contrast, in high-prevalence regions such as sub-Saharan Africa, measures of prompt switching between first-line drugs and combination therapies will be quite robust against rapid increases and prolonged maintenance of drug resistance (*Flegg et al., 2013*).

## Methods
### Transmission dynamics
Rather than following the infected vector populations, transmission potential is given by a fixed contact rate, which represents the contact rate per host at which a mosquito bites a donor host, gets infected, survives the sporogonic period, and transmits to a recipient host. This contact rate is uniform across all host classes. Hosts may harbor 0 to $n_{max}$ strains of parasites. Those with $MOI > 0$ will be able to infect mosquitoes. However, a strain from the donor does not guarantee its successful infection in a recipient. Instead, the infections will not result if the host has reached its carrying capacity of $n_{max}$ strains, at which they cannot harbor more infections, or if the host has encountered and acquired the specific immunity to the strain (*Figure 1A*). In these cases, the MOI in the host remains constant. Otherwise, infection will result, and MOI will increase by 1.

### Calculating MOI and parasite prevalence
A major assumption that links host and parasite populations is that the number of infections in an individual host (i.e., MOI) at any time follows some prespecified distribution. To reduce the number of parameters and simplify the model, a Poisson distribution was used for MOI within a given $G$ and treatment class. This assumption allows us to directly calculate the prevalence (i.e., the fraction of individuals carrying at least one infection) in a given $G$ class $i = 0, 1, 2$ and treatment class $j = U, D$ as

$$I_{i,j} = 1 - \exp(-r_{i,j}) \tag{2}$$

where $r$ is the mean MOI of the class and is equal to

$$r_{i,j} = \frac{PW_{i,j} + PR_{i,j}}{H_{i,j}} \tag{3}$$

where $PW_{i,j}$ and $PR_{i,j}$ are the numbers of wild-type ($W$) and resistant ($R$) infections circulating in the host class at a given time, and are determined from the system of mechanistic differential equations (**Figure 1—figure supplement 1B**). $H_{i,j}$ is the number of hosts in the class at a given time and is similarly determined by the ODE system (**Figure 1—figure supplement 1A**).

One justification for using a Poisson distribution for MOI is a reduction in complexity given a lack of knowledge from empirical data; however, the model can be extended to include an implicit clustering if the Poisson distribution is replaced by an NMD.

Finally, the population-level prevalence is thus the summation of prevalence in individual host classes,

$$I = \sum_{i,j} I_{i,j} \times (H_{i,j} / \sum H_{i,j}) \tag{4}$$

## MOI-dependent versus MOI-independent rates

The macroparasite modeling approach also impacts how transition rates are calculated, which is different from typical SIR models. Some transition rates of host classes in the ODE system are dependent on the number of parasite infections (i.e., MOI), whereas some are independent of MOI. For example, host natural death rate ($H_{ij}\alpha$) is MOI-independent because the rate itself need not be weighted by an additional factor related to MOI. Accordingly, parasite death rate due to host natural death is ($PW_{ij} + PR_{ij})\alpha$. Alternatively, host drug treatment rate depends on MOI. The value of this rate is explicitly equal to

$$(G_{i,U} \to G_{i,D}) = H_{i,U}(t) \sum_{k=1|i,U}^{\infty} k \cdot p(k) d \tag{5}$$

where each $k$ is the number of infections in a given host, $p(k)$ is the fraction of hosts having $k$ infections, and $d$ is the fixed treatment rate upon experiencing symptoms. The reason the second term is necessary is to count each separate infection as a different chance to experience symptoms. Given that this term is equal to $r_{i,U} = \frac{PW_{i,U}(t) + PR_{i,U}(t)}{H_{i,U}(t)}$, we get that

$$(G_{i,U} \to G_{i,D}) = (PW_{i,U}(t) + PR_{i,U}(t))d \tag{6}$$

Thus, the movement rates for parasites from untreated classes to drug-treated classes need to consider the host movement rates as well as the number of parasites that are 'carried' by the hosts. Using resistant parasites as an example,

$$\begin{aligned}(PR_{i,U} \to PR_{i,D}) &= H_{i,U}(t) \sum_{k=1|i,U}^{\infty} k \cdot d \cdot k \cdot p(k) \\ &= H_{i,U}(t) E(k^2) d \end{aligned} \tag{7}$$

where $E(k^2)$ refers to the expectation value of the square of the MOI distribution. Given that this expectation value can be written as $var(k) + (E(k))^2$, given the Poisson assumption (which implies that $var(k)$ and $E(k)$ are equal), we finally get an overall rate of

$$(PR_{i,U} \to PR_{i,D}) = PR_{i,U}(t)(1 + r_{i,U}^{PR})d \tag{8}$$

where $r_{i,U}^{PR}$ is the mean MOI ($E(k)$) of resistant parasites in the $G_{i,U}$ class.

## Cost of resistance and contributions of wild-type and resistant parasites to transmission

Also calculated using Poisson statistics are the contributions of the two parasite genotypes to transmission originating from a host in a given $G$ class. These contributions are dependent on two fixed cost parameters: the fitness cost to transmission associated with resistance in the absence of sensitive parasites ($s_{single}$, for single-genotype), and the fitness cost to transmission associated with resistance due to competition with wild-type parasites present in the same host ($s_{mixed}$, for mixed-genotype). Parasite density is assumed to be regulated by similar resources within a host (e.g., red blood cells) regardless of MOI. Thus, each strain has a reduced transmissibility when MOI > 1. For wild-type-only infections of MOI = $k$, each strain has transmissibility of $1/k$; for resistant-only infections, each strain has transmissibility of $1/k \cdot (1 - s_{single})$; for mixed-genotype infections, if there are $m$ wild-type strains and $n$ resistant strains, transmission from $n$ resistant strains is $\frac{n}{m+n}(1 - s_{mixed})$, while transmission from $m$ wild-type strains is $\frac{m}{m+n} + \frac{n}{m+n}s_{mixed}$ assuming wild-type strains outcompete resistant strains in growth rates and reach a higher cumulative density during the infective period.

Based on these assumptions, we then calculate transmissibility contributions at the population level from wild-type strains in purely wild-type infections ($\phi_{WS,ij}$), wild-type strains in mixed-genotype infections ($\phi_{WM,ij}$), resistant strains in purely resistant infections ($\phi_{RS,ij}$), and resistant strains in mixed-genotype infections ($\phi_{RM,ij}$). Details on how these terms were calculated using Poisson statistics are provided in Appendix 1. The total contributions to transmissibility from resistant and sensitive parasites at a given time step are then

$$\Omega_{W,ij} = \phi_{WS,ij} + \phi_{WM,ij} + (\phi_{RM,ij})(s_{mixed}) \tag{9}$$

$$\Omega_{R,ij} = (\phi_{RS,ij})(1 - s_{single}) + (\phi_{RM,ij})(1 - s_{mixed}) \tag{10}$$

These contributions can then be used to determine the realized transmission rates given a transmission potential, as shown in Appendix 1.

## The process of immunity loss

A significant challenge in developing the model is to describe a function for immunity loss for a given class. We adopted the classic equations for the dynamics of acquired immunity boosted by exposure to infection (Eq. 2.5 from *Aron, 1983*). This gives the following immunity loss rate from a higher generalized immunity class to a lower one:

$$(G_{i,j} \rightarrow G_{i-1,j}) = h_{i,j} \frac{\exp(-\frac{h_{i,j}}{\Lambda})}{1 - \exp(-\frac{h_{i,j}}{\Lambda})} \tag{11}$$

In this case, $h_{i,j}$ is the sum of the inoculation rate and host death rate for the $G_{i,j}$ class and is determined mechanistically, and $\Lambda$ is a fixed immunity loss rate parameter with dimensions of 1/[*time*]. The second factor in the equation represents the failure of boosting, that is, the probability that an individual is infected after the period of immunity has ended given that they were not infected within the immune state (*Aron and May, 1982*).

## Drug treatment and resistance invasion

Given each parameter set, we ran the ODE model six times until equilibrium with the following genotypic compositions: (1) wild-type-only scenario with no drug treatment; (2) wild-type-only scenario with 63.2% drug treatment (0.05 daily treatment rate); (3) wild-type-only scenario with 98.2% drug treatment (0.2 daily treatment rate); (4) resistant-only scenario with no drug treatment; (5) resistance invasion with 63.2% drug treatment; and (6) resistance invasion with 98.2% drug treatment. Runs 1–4 start with all hosts in $G_{0,U}$ compartment and 10 parasites. Runs 5 and 6 (resistance invasion) start from the equilibrium state of 2 and 3, with 10 resistant parasites introduced. We then followed the ODE dynamics till the next equilibrium.

**Table 1.** Empirical ranges of transmission potential ($k_0$) of different continents.
$k_0 = C \times g$, where $g$ is set at 0.08.

| Continent | $C$ | $k_0$ | Source |
|---|---|---|---|
| Africa | 0.54–16.2 | 0.043–1.3 | *Dietz et al., 1974*; *Garrett-Jones and Shidrawi, 1969*; *Afrane et al., 2008* |
| Asia | 0.014–6.5 | 0.0011–0.52 | *Rattanarithikul et al., 1996*; *Rosenberg et al., 1990*; *Toma et al., 2002*; *Vythilingam et al., 2003*; *Zhou et al., 2010*; *Gunasekaran et al., 2014*; *Edalat et al., 2016* |
| Oceania | 1.60–9.64 | 0.13–0.77 | *Graves et al., 1990* |
| South America | 0.88–5.53 | 0.070–0.44 | *Rubio-Palis, 1994*; *Zimmerman et al., 2022* |

## Sources of empirical parameters, measures and policies, and regression analysis

We define transmission potential ($k_0$) as effective contact rate via vectors, consistent with a recent immune-structured SIS model that does not explicitly model vector dynamics (*de Roos et al., 2023*). Specifically, it is the product of vectorial capacity ($C$) (*Garrett-Jones, 1964*) and the maximum transmissibility between host and mosquito in one transmission cycle ($g$) (*Dietz et al., 1974*). The transmissibility, $g$, has two components: infectivity of malaria patients to mosquitoes and transmissibility from infected mosquitoes to humans. The infectivity of falciparum malaria patients to mosquitoes was estimated to be around 0.1 from multiple studies (*Timinao et al., 2021*; *Coleman et al., 2004*). Transmissibility from infected mosquitoes to naïve hosts is around 0.8 for sporozoites density higher than 1000 per mosquito (*Churcher et al., 2005*). Thus, we set $g$ to be 0.08 in calculating empirical $k_0$. Empirical estimates of vectorial capacities were compiled from all known studies of different countries. Ranges of vectorial capacities, $C$, reported in *Table 1* were calculated by summing $C$ of *P. falciparum* from all the local vectors. Transmission potential ($k_0$) for each continent was then obtained by the product of $C$ and $g$.

Empirical strain diversities were calculated by the local estimates of *var* diversity divided by the number of unique non-shared types per strain for each region (*Table 2*). Note that *var* diversity in Asia from genomic sequencing was only available for two countries: Thailand and Iran. The variation in strain diversity in Asia might be underestimated.

We acquired resistance marker *pfcrt* 76T frequencies from the Worldwide Antimalarial Resistance Network (WWARN). The website obtained resistant frequencies from 587 studies between 2001 and 2022 with specific curation methodologies. We then extracted geographic sampling locations from the database, and extracted *Pf* prevalence data estimated from 2- to 10-year-old children from Malaria Atlas Project. The Malaria Atlas Project does not have predicted prevalence before 2000, while the change in first-line antimalarial drugs started around early 2000 in most African countries. We, therefore, restricted our empirical comparisons of equilibrium levels of resistance and prevalence to studies that conducted surveys between 1990 and 2000 and used estimated prevalence from the year 2000 as the proxy for this sampling period.

**Table 2.** Empirical ranges of strain diversity of different continents.
$u$, unique non-shared types per strain. $D_{var}$ is the Chao1 index (*Chao, 1984*) estimated from local sampling.

| Continent | $D_{var}$ | $u$ | nstrain | Source |
|---|---|---|---|---|
| Africa | 3712–20,000 | 50 | 74.24–400 | *Chen et al., 2011*; *Day et al., 2017*; *Ruybal-Pesántez et al., 2022* |
| Asia | 1100–1700 | 25 | 44–68 | *Tonkin-Hill et al., 2018* |
| Oceania | 290–1094 | 20 | 14.5–54.7 | *Barry et al., 2007*; *Tessema et al., 2015* |
| South America | 113–351 | 25 | 4.52–14.04 | *Albrecht et al., 2010*; *Rougeron et al., 2017* |

**Table 3.** Source of drug policy data.

| Country | Citations |
| --- | --- |
| Kenya | *Hemming-Schroeder et al., 2018* |
| Ghana | *Flegg et al., 2013* |
| Cambodia | *Delacollette et al., 2009* |
| Thailand | *Delacollette et al., 2009*; *Rasmussen et al., 2022* |
| Papua New Guinea | *Nsanzabana et al., 2010* |
| Brazil | *Gama et al., 2009* |

Studies with a host sampling size of less than 20 were excluded. Data sources on drug usage and policies for different countries are summarized in *Table 3*.

The relationship between prevalence and resistant frequency was investigated using beta regression because both the explanatory variable and response variable are proportions, restricted to the unit interval (0,1) (*Ferrari and Cribari-Neto, 2004*; *Simas et al., 2010*). Thus, the proper distribution of the response variable (here, resistant prevalence) should be a beta distribution with a mean and precision parameter. Since resistant frequency also has extremes of 0 and 1, we transformed the frequency data to restrict its range between 0 and 1 first so that beta regression still applies,

$$freq_{adj} = (freq \cdot (n-1) + 0.5)/n \tag{12}$$

where $n$ is the sample size (*Smithson and Verkuilen, 2006*). We then used *betareg* function from R package 4.2.1 *betareg* 3.1–4 to perform the regression (*Cribari-Neto and Zeileis, 2010*).

## Acknowledgements

We thank Mercedes Pascual for valuable suggestions on earlier versions of the model, and Karen Day and Kathryn Tiedje for their helpful discussions and feedback on this work. We thank insightful comments from two anonymous reviewers that helped improve the work significantly. We appreciate the support of Information Technology at Purdue University through the computational resources of the Bell Community Cluster. This work was partially supported by the joint NIH-NSF-NIFA Ecology and Evolution of Infectious Disease award R01-AI149779 to Karen Day and Mercedes Pascual.

## Additional information

### Funding

| Funder | Grant reference number | Author |
| --- | --- | --- |
| Fogarty International Center | R01-AI149779 | Frédéric Labbé |

The funders had no role in study design, data collection and interpretation, or the decision to submit the work for publication.

### Author contributions

Qixin He, Conceptualization, Resources, Data curation, Formal analysis, Supervision, Validation, Visualization, Methodology, Writing – original draft, Project administration, Writing – review and editing; John K Chaillet, Conceptualization, Data curation, Formal analysis, Validation, Visualization, Methodology, Writing – original draft, Writing – review and editing; Frédéric Labbé, Visualization, Methodology, Writing – original draft, Writing – review and editing

### Author ORCIDs

Qixin He (iD) https://orcid.org/0000-0003-1696-8203

John K Chaillet ⓘ http://orcid.org/0000-0002-6156-4649
Frédéric Labbé ⓘ https://orcid.org/0000-0002-4064-2361

Reviewer #1 (Public Review): https://doi.org/10.7554/eLife.90888.3.sa1
Reviewer #2 (Public Review): https://doi.org/10.7554/eLife.90888.3.sa2
Author Response https://doi.org/10.7554/eLife.90888.3.sa3

## Additional files

### Supplementary files
• MDAR checklist

### Data availability
All the ODE codes, numerically-simulated data, empirical data, and analyzing scripts are publicly available at https://github.itap.purdue.edu/HeLab/MalariaResistance (copy archived at *Qixin and Chaillet, 2023*).

The following previously published datasets were used:

| Author(s) | Year | Dataset title | Dataset URL | Database and Identifier |
|---|---|---|---|---|
| Infectious Diseases Data Observatory (IDDO) | 2015 | The ACT Partner Drug Molecular Surveyor | https://www.iddo.org/wwarn/tracking-resistance/act-partner-drug-molecular-surveyor | Infectious Diseases Data Observatory, act-partner-drug-molecular-surveyor |
| The Malaria Atlas Project | 2023 | *Plasmodium falciparum* Infection Prevalence | https://data.malariaatlas.org/trends?metricGroup=Malaria&metricSubcategory=Pf&metricType=rate&metricName=PR&year=2020&geographicLevel=admin0 | The Malaria Atlas Project, https://data.malariaatlas.org/trends?metricGroup=Malaria&metricSubcategory=Pf&metricType=rate&metricName=PR&year=2020&geographicLevel=admin0 |

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

## Appendix 1

### Basic model structure

The primary structure of the macroparasitic model is composed of three submodels: (i) the number of hosts in generalized immunity classes, (ii) circulating infections (parasites) in host GI classes, and (iii) the dynamics of immune memory of host GI classes (*Figure 1—figure supplement 1*; list of parameters: *Appendix 1—table 1*, 'Final implementation'). These submodels are interconnected: the infectivity of new parasites is determined by the accumulated immune memory in the different host classes; some of the transition rates of host classes are dependent on the number of parasite infections (i.e., MOI), whereas some are independent of MOI; while the transition rates of resistant or wild-type parasites between untreated host classes are identical, they experience vastly different survival rates in treated classes.

The host submodel divides human hosts into three generalized immunity classes, dubbed class $G_0$, class $G_1$, and class $G_2$. These immunity classes are defined by the extent to which generalized immunity causes them to experience symptoms, as well as the severity of those symptoms. Each immunity class is further separated into drug-treated ($D$) and untreated ($U$) classes. The populations of hosts in the different immune classes are denoted by $H_{0,U}$, $H_{0,D}$, $H_{1,U}$, $H_{1,D}$, $H_{2,U}$, and $H_{2,D}$. The total host population is referred to as $H$, and the total number of extant infections in the population is $P$.

Parasites are categorized according to their genotype and associated host classes, in which $PW$ and $PR$ denote wild-type and resistant parasite populations, respectively. With the associated host generalized immunity class structure, the parasite classes are further subdivided into $PW_{0,U}$, $PW_{0,D}$, $PW_{1,U}$, $PW_{1,D}$, $PW_{2,U}$, $PW_{2,D}$, $PR_{0,U}$, $PR_{0,D}$, $PR_{1,U}$, $PR_{1,D}$, $PR_{2,U}$, and $PR_{2,D}$.

We assume parasites follow a Poisson distribution within each host class. Therefore, the prevalence (i.e., the fraction of individuals carrying at least one infection) in a given $G$ class $i = 0, 1, 2$ and treatment class $j = U, D$ is,

$$I_{i,j} = 1 - \exp(-r_{i,j}) \tag{13}$$

where $r$ is the mean MOI (parasite-to-host ratio) of the class and is equal to

$$r_{i,j} = \frac{PW_{i,j} + PR_{i,j}}{H_{i,j}} \tag{14}$$

Thus, the overall population-level prevalence is

$$I = \sum_{i,j} I_{i,j} \times (H_{i,j} / \sum H_{i,j}) \tag{15}$$

Further using the Poisson assumption, the proportions of hosts in each class that have no sensitive parasites are

$$p_{0_{W,ij}} = e^{-r_{i,j} \frac{PW_{i,j}}{PW_{i,j} + PR_{i,j}}} \tag{16}$$

The equivalent values for the proportions of hosts in each class that have no resistant parasites are

$$p_{0_{R,ij}} = e^{-r_{i,j} \frac{PR_{i,j}}{PW_{i,j} + PR_{i,j}}} \tag{17}$$

Given these proportions, we can calculate the transmissibility contributions from wild-type strains in purely wild-type infections ($\phi_{WS,ij}$), wild-type strains in mixed-genotype infections ($\phi_{WM,ij}$), resistant strains in purely resistant infections ($\phi_{RS,ij}$), and resistant strains in mixed-genotype infections ($\phi_{RM,ij}$):

$$\phi_{WS,ij} = (1 - p_{0_{W,ij}}) p_{0_{R,ij}} \tag{18}$$

$$\phi_{WM,ij} = (1 - p_{0_{W,ij}})(1 - p_{0_{R,ij}}) \frac{PW_{i,j}}{PW_{i,j} + PR_{i,j}} \tag{19}$$

$$\phi_{RS,ij} = (1 - p_{0_{R,ij}})p_{0_{W,ij}} \tag{20}$$

$$\phi_{RM,ij} = (1 - p_{0_{W,ij}})(1 - p_{0_{R,ij}})\frac{PR_{i,j}}{PW_{i,j} + PR_{i,j}} \tag{21}$$

The total converted contributions of $W$ and $R$ to transmissibility are therefore

$$\Omega_{W,ij} = \phi_{WS,ij} + \phi_{WM,ij} + (\phi_{RM,ij})(s_{mixed}) \tag{22}$$

$$\Omega_{R,ij} = (\phi_{RS,ij})(1 - s_{single}) + (\phi_{RM,ij})(1 - s_{mixed}) \tag{23}$$

where $s_{single}$ is the fitness cost to transmission associated with resistance in the absence of wild-type parasites and $s_{mixed}$ is the fitness cost to transmission associated with resistance due to competition with wild-type parasites present in the same host. Note that it is assumed that any loss in transmissibility to the resistant parasites due to the mixed cost is recovered by an increase in transmissibility in the cooccurring wild-type parasites.

For the fixed biting rate parameter $b$, the transmissibilities of wild-type and resistant parasites on the whole are therefore

$$\begin{cases} B_{PW} = \frac{b}{H} \sum_{i,j} \Omega_{W,ij} H_{i,j} \\ B_{PR} = \frac{b}{H} \sum_{i,j} \Omega_{R,ij} H_{i,j} \end{cases} \tag{24}$$

The per capita biting rate for untreated hosts is simply

$$B_U = B_{PW} + B_{PR} \tag{25}$$

It is assumed that drug-treated hosts can only be infected by resistant parasites, so the effective per capita biting rate for drug-treated hosts is

$$B_D = B_{PR} \tag{26}$$

A unique feature of the first generalized immunity class ($i = 0$) in our model is that there is a significant death rate from the disease, termed $\mu_{0,U}$ or $\mu_{0,D}$. Once a host has moved from the first to the second immune stage, their chance of experiencing symptoms is determined by their specific immunity. Finally, in the third immune class, the hosts can only experience symptoms when a sufficiently distinct migrant strain enters the population (at probability $\omega$).

Given the generalized immunity tracked in each host class, the degree of specific immunity per class is determined as the infectivity of new infection to that class,

$$\eta_i = (1 - \frac{1}{n_{strains}})^{\nu_i} \tag{27}$$

where $n_{strains}$ represents the number of strains in the local population and $\nu$ for a given class $i$ refers to the generalized immunity of that class (in terms of the number of infections experienced by an average individual currently in class $i$). $\eta_i$ therefore calculates the probability that the host in that $G$ class has not seen a particular strain.

In the following sections, we list how transition rates are calculated in the submodels. Each rate is uniquely marked with the submodel letter and a number, as notated on *Figure 1—figure supplement 1*.

## Host submodel
The transition rates between compartments in the host model use the following rates:
**Birth**

$$\begin{cases} A1 = \delta \end{cases} \tag{28}$$

where $\delta$ is the constant birth rate (into $G_{0_U}$).
**Malarial death**

$$\begin{cases} A2 = \mu_{0,U} H_{0,U} \\ A3 = \mu_{0,D} H_{0,D} \end{cases} \tag{29}$$

**Natural death**

$$\begin{cases} A4 = \alpha H_{0,U} \\ A5 = \alpha H_{0,D} \\ A6 = \alpha H_{1,U} \\ A7 = \alpha H_{1,D} \\ A8 = \alpha H_{2,U} \\ A9 = \alpha H_{2,D} \end{cases} \tag{30}$$

where $\alpha$ is the non-disease death rate for the host population.

**Drug treatment of symptomatic individuals (MOI-dependent)**

$$\begin{cases} A10 = r_{0,U} d_0 H_{0,U} \\ A11 = r_{1,U} d_1 H_{1,U} \\ A12 = r_{2,U} d_2 \dfrac{\omega}{n_{strains}} H_{2,U} \end{cases} \tag{31}$$

where $d_i$ is the daily treatment rate of hosts in a given class who are currently experiencing symptoms.

**Loss of drug effectiveness**

$$\begin{cases} A13 = \dfrac{H_{0,D}}{\tau} \\ A14 = \dfrac{H_{1,D}}{\tau} \\ A15 = \dfrac{H_{2,D}}{\tau} \end{cases} \tag{32}$$

where $\tau$ is the period of drug effectiveness.

**Gain of generalized immunity**

$$\begin{cases} A16 = (B_U(1 - \dfrac{r_{0,U}}{K})\rho_1(1 - \eta_0) + \rho_1 r_{0,U} \mu_{PW_U}) H_{0,U} \\ A17 = (B_D(1 - \dfrac{r_{0,D}}{K})\rho_1(1 - \eta_0) + \rho_1 r_{0,D} \mu_{PR_D}) H_{0,D} \\ A18 = (B_U(1 - \dfrac{r_{1,U}}{K})\rho_2(1 - \eta_1) + \rho_2 r_{1,U} \mu_{PW_U}) H_{1,U} \\ A19 = (B_D(1 - \dfrac{r_{1,D}}{K})\rho_2(1 - \eta_1) + \rho_2 r_{1,D} \mu_{PR_D}) H_{1,D} \end{cases} \tag{33}$$

where $K$ is the per-host carrying capacity for infections, $\rho_1$ is the rate of gaining generalized immunity from class 0 to class 1, $\rho_2$ is the rate of gain of generalized immunity from class 1 to class 2, and $\mu_{PW_U}$, etc., are the parasite clearance rates in different host treatment categories. The first part of the rate indicates that the host receives a previous-seen infection, so it will not result in a new infection, instead, the $G$ is boosted by 1. The second part of the rate indicates that as current infections in the host class are being cleared, $G$ is also boosted.

**Loss of generalized immunity**

$$\begin{cases} A20 = H_{1,U}h_{1,U}\dfrac{\exp(-\dfrac{h_{1,U}}{\Lambda})}{1 - \exp(-\dfrac{h_{1,U}}{\Lambda})} \\[3mm] A21 = H_{1,D}h_{1,D}\dfrac{\exp(-\dfrac{h_{1,D}}{\Lambda})}{1 - \exp(-\dfrac{h_{1,D}}{\Lambda})} \\[3mm] A22 = H_{2,U}h_{2,U}\dfrac{\exp(-\dfrac{h_{2,U}}{\Lambda})}{1 - \exp(-\dfrac{h_{2,U}}{\Lambda})} \\[3mm] A23 = H_{2,D}h_{2,D}\dfrac{\exp(-\dfrac{h_{2,D}}{\Lambda})}{1 - \exp(-\dfrac{h_{2,D}}{\Lambda})} \end{cases} \tag{34}$$

$$\begin{cases} h_{1,U} = (B_U(1 - \dfrac{r_{1,U}}{K})) + \alpha \\[3mm] h_{1,D} = (B_D(1 - \dfrac{r_{1,D}}{K})) + \alpha \\[3mm] h_{2,U} = (B_U(1 - \dfrac{r_{2,U}}{K})) + \alpha \\[3mm] h_{2,D} = (B_D(1 - \dfrac{r_{2,D}}{K})) + \alpha \end{cases} \tag{35}$$

where *Lambda* is the immunity loss rate. The immunity loss rates follow *Aron, 1983* formulation. Therefore, ODEs for the host submodel are

$$\begin{cases} \dfrac{dH_{0,U}}{dt} = (A1) + (A20) + (A13) - (A10) - (A16) - (A2) - (A4) \\[3mm] \dfrac{dH_{0,D}}{dt} = (A21) + (A10) - (A13) - (A17) - (A3) - (A5) \\[3mm] \dfrac{dH_{1,U}}{dt} = (A22) + (A14) + (A16) - (A11) - (A18) - (A20) - (A6) \\[3mm] \dfrac{dH_{1,D}}{dt} = (A23) + (A11) + (A17) - (A14) - (A19) - (A21) - (A7) \\[3mm] \dfrac{dH_{2,U}}{dt} = (A18) + (A15) - (A12) - (A22) - (A8) \\[3mm] \dfrac{dH_{2,D}}{dt} = (A19) + (A12) - (A15) - (A23) - (A9) \end{cases} \tag{36}$$

## Parasite submodel

For the parasite submodel, it is necessary to define a new set of rates, which are mostly variations on the rates from the host submodel:

**Infection via vector**

$$\begin{cases} BW1 = B_{PW}H_{0,U}\eta_0(1 - \dfrac{r_{0,U}}{K}) \\ BW2 = B_{PW}H_{1,U}\eta_1(1 - \dfrac{r_{1,U}}{K}) \\ BW3 = B_{PW}H_{2,U}\eta_2(1 - \dfrac{r_{2,U}}{K}) \end{cases} \tag{37}$$

**Transition due to host getting treated: MOI-dependent**

$$\begin{cases} BW4 = (1 + r_{0,U})d_0 PW_{0,U} \\ BW5 = (1 + r_{1,U})d_1 PW_{1,U} \\ BW6 = (1 + r_{2,U})d_2 \dfrac{\omega}{n_{strains}} PW_{2,U} \end{cases} \tag{38}$$

**Parasite survival of drug treatment**

$$\begin{cases} BW7 = \dfrac{PW_{0,D}}{\tau} \\ BW8 = \dfrac{PW_{1,D}}{\tau} \\ BW9 = \dfrac{PW_{2,D}}{\tau} \end{cases} \tag{39}$$

**Transition due to host gain of immunity**

$$\begin{cases} BW10 = B_U(1 - \dfrac{r_{0,U}}{K})\rho_1(1 - \eta_0) + \rho_1(1 + r_{0,U})\mu_{PW_U}PW_{0,U} \\ BW11 = B_D(1 - \dfrac{r_{0,D}}{K})\rho_1(1 - \eta_0) + \rho_1(1 + r_{0,D})\mu_{PR_D}PW_{0,D} \\ BW12 = B_U(1 - \dfrac{r_{1,U}}{K})\rho_2(1 - \eta_1) + \rho_2(1 + r_{1,U})\mu_{PW_U}PW_{1,U} \\ BW13 = B_D(1 - \dfrac{r_{1,D}}{K})\rho_2(1 - \eta_1) + \rho_2(1 + r_{1,D})\mu_{PR_D}PW_{1,D} \end{cases} \tag{40}$$

**Transition due to host loss of immunity**

$$\begin{cases} BW14 = (H_{1,U}h_{1,U}\dfrac{\exp(-\dfrac{h_{1,U}}{\Lambda})}{1 - \exp(-\dfrac{h_{1,U}}{\Lambda})})PW_{1,U} \\ BW15 = (H_{1,D}h_{1,D}\dfrac{\exp(-\dfrac{h_{1,D}}{\Lambda})}{1 - \exp(-\dfrac{h_{1,D}}{\Lambda})})PW_{1,D} \\ BW16 = (H_{2,U}h_{2,U}\dfrac{\exp(-\dfrac{h_{2,U}}{\Lambda})}{1 - \exp(-\dfrac{h_{2,U}}{\Lambda})})PW_{2,U} \\ BW17 = (H_{2,D}h_{2,D}\dfrac{\exp(-\dfrac{h_{2,D}}{\Lambda})}{1 - \exp(-\dfrac{h_{2,D}}{\Lambda})})PW_{2,D} \end{cases} \tag{41}$$

**Parasites removed due to malarial death**

$$\begin{cases} BW18 = (1 + r_{0,U})\mu_{PW_U}\mu_{0,U}PW_{0,U} \\ BW19 = (1 + r_{0,D})\mu_{PR_D}\mu_{0,D}PW_{0,D} \end{cases} \tag{42}$$

**Parasite death due to immunity or drug clearance in different host environments**

$$\begin{cases} BW20 = \mu_{PW_U}PW_{0,U} \\ BW21 = \mu_{PW_D}PW_{0,D} \\ BW22 = \mu_{PW_U}PW_{1,U} \\ BW23 = \mu_{PW_D}PW_{1,D} \\ BW24 = \mu_{PW_U}PW_{2,U} \\ BW25 = \mu_{PW_D}PW_{2,D} \end{cases} \quad (43)$$

**Parasite death due to natural host death**

$$\begin{cases} BW26 = \alpha PW_{0,U} \\ BW27 = \alpha PW_{0,D} \\ BW28 = \alpha PW_{1,U} \\ BW29 = \alpha PW_{1,D} \\ BW30 = \alpha PW_{2,U} \\ BW31 = \alpha PW_{2,D} \end{cases} \quad (44)$$

These rates can be thought of as accounting separately for new infections versus concurrent infections. The terms $(1 + r_{i,j})$ are to account for the MOI dependence explained and derived in the main text.

The ODEs for sensitive (wild-type) strains in the parasite model are as follows. (Note that hosts in $G_2$ can still be infected by local strains in the model, even if they do not experience symptoms.):

$$\begin{cases} \dfrac{dPR_{0,U}}{dt} = (BW1) + (BW7) + (BW14) - (BW4) - (BW10) - (BW18) - (BW20) - (BW26) \\[2mm] \dfrac{dPR_{0,D}}{dt} = (BW4) + (BW15) - (BW7) - (BW11) - (BW19) - (BW21) - (BW27) \\[2mm] \dfrac{dPR_{1,U}}{dt} = (BW2) + (BW8) + (BW10) + (BW16) - (BW5) - (BW12) - (BW14) - (BW22) - (BW28) \\[2mm] \dfrac{dPR_{1,D}}{dt} = (BW5) + (BW11) + (BW17) - (BW8) - (BW13) - (BW15) - (BW23) - (BW29) \\[2mm] \dfrac{dPR_{2,U}}{dt} = (BW3) + (BW9) + (BW12) - (BW6) - (BW16) - (BW24) - (BW30) \\[2mm] \dfrac{dPR_{2,D}}{dt} = (BW6) + (BW13) - (BW9) - (BW17) - (BW25) - (BW31) \end{cases} \quad (45)$$

A similar set of rates is used for the formulation of the ODEs for the resistant parasite classes:
**Infection via vector**

$$\begin{cases} BR1 = B_{PR}H_{0,U}\eta_0(1 - \dfrac{r_{0,U}}{K}) \\ BR2 = B_{PR}H_{1,U}\eta_1(1 - \dfrac{r_{1,U}}{K}) \\ BR3 = B_{PR}H_{2,U}\eta_2(1 - \dfrac{r_{2,U}}{K}) \\ BR4 = B_{PR}H_{0,D}\eta_0(1 - \dfrac{r_{0,D}}{K}) \\ BR5 = B_{PR}H_{1,D}\eta_1(1 - \dfrac{r_{1,D}}{K}) \\ BR6 = B_{PR}H_{2,D}\eta_2(1 - \dfrac{r_{2,D}}{K}) \end{cases} \quad (46)$$

**Transition due to treatment**

$$
\begin{cases}
BR7 = (1 + r_{0,U})d_0 PR_{0,U} \\
BR8 = (1 + r_{1,U})d_1 PR_{1,U} \\
BR9 = (1 + r_{2,U})d_2 \dfrac{\omega}{n_{strains}} PR_{2,U}
\end{cases}
\tag{47}
$$

**Loss of drug effectiveness**

$$
\begin{cases}
BR10 = \dfrac{PR_{0,D}}{\tau} \\
BR11 = \dfrac{PR_{1,D}}{\tau} \\
BR12 = \dfrac{PR_{2,D}}{\tau}
\end{cases}
\tag{48}
$$

**Transition due to host gain of immunity**

$$
\begin{cases}
BR13 = B_U(1 - \dfrac{r_{0,U}}{K})\rho_1(1 - \eta_0) + \rho_1(1 + r_{0,U})\mu_{PW_U} PR_{0,U} \\
BR14 = B_D(1 - \dfrac{r_{0,D}}{K})\rho_1(1 - \eta_0) + \rho_1(1 + r_{0,D})\mu_{PR_D} PR_{0,D} \\
BR15 = B_U(1 - \dfrac{r_{1,U}}{K})\rho_2(1 - \eta_1) + \rho_2(1 + r_{1,U})\mu_{PW_U} PR_{1,U} \\
BR16 = B_D(1 - \dfrac{r_{1,D}}{K})\rho_2(1 - \eta_1) + \rho_2(1 + r_{1,D})\mu_{PR_D} PR_{1,D}
\end{cases}
\tag{49}
$$

**Transition due to host loss of immunity**

$$
\begin{cases}
BR17 = (H_{1,U} h_{1,U} \dfrac{\exp(-\dfrac{h_{1,U}}{\Lambda})}{1 - \exp(-\dfrac{h_{1,U}}{\Lambda})})PR_{1,U} \\[2em]
BR18 = (H_{1,D} h_{1,D} \dfrac{\exp(-\dfrac{h_{1,D}}{\Lambda})}{1 - \exp(-\dfrac{h_{1,D}}{\Lambda})})PR_{1,D} \\[2em]
BR19 = (H_{2,U} h_{2,U} \dfrac{\exp(-\dfrac{h_{2,U}}{\Lambda})}{1 - \exp(-\dfrac{h_{2,U}}{\Lambda})})PR_{2,U} \\[2em]
BR20 = (H_{2,D} h_{2,D} \dfrac{\exp(-\dfrac{h_{2,D}}{\Lambda})}{1 - \exp(-\dfrac{h_{2,D}}{\Lambda})})PR_{2,D}
\end{cases}
\tag{50}
$$

**Transition due to malarial death**

$$
\begin{cases}
BR21 = (1 + r_{0,U})\mu_{PW_U}\mu_{0,U} PR_{0,U} \\
BR22 = (1 + r_{0,D})\mu_{PR_D}\mu_{0,D} PR_{0,D}
\end{cases}
\tag{51}
$$

**Parasite death in different host environments**

$$
\begin{cases}
BR23 = \mu_{PR_U} PR_{0,U} \\
BR24 = \mu_{PR_D} PR_{0,D} \\
BR25 = \mu_{PR_U} PR_{1,U} \\
BR26 = \mu_{PR_D} PR_{1,D} \\
BR27 = \mu_{PR_U} PR_{2,U} \\
BR28 = \mu_{PR_D} PR_{2,D}
\end{cases}
\tag{52}
$$

**Transition due to natural host death**

$$
\begin{cases}
BR29 = \alpha PR_{0,U} \\
BR30 = \alpha PR_{0,D} \\
BR31 = \alpha PR_{1,U} \\
BR32 = \alpha PR_{1,D} \\
BR33 = \alpha PR_{2,U} \\
BR34 = \alpha PR_{2,D}
\end{cases}
\tag{53}
$$

The corresponding set of ODEs for resistant parasite populations is therefore

$$
\begin{cases}
\dfrac{dPR_{0,U}}{dt} = (BR1) + (BR10) + (BR17) - (BR7) - (BR13) - (BR21) - (BR23) - (BR29) \\[2mm]
\dfrac{dPR_{0,D}}{dt} = (BR4) + (BR7) + (BR18) - (BR10) - (BR14) - (BR22) - (BR24) - (BR30) \\[2mm]
\dfrac{dPR_{1,U}}{dt} = (BR2) + (BR11) + (BR13) + (BR19) - (BR8) - (BR15) - (BR17) - (BR25) - (BR31) \\[2mm]
\dfrac{dPR_{1,D}}{dt} = (BR5) + (BR8) + (BR14) + (BR20) - (BR11) - (BR16) - (BR18) - (BR26) - (BR32) \\[2mm]
\dfrac{dPR_{2,U}}{dt} = (BR3) + (BR12) + (BR15) - (BR9) - (BR19) - (BR27) - (BR33) \\[2mm]
\dfrac{dPR_{2,D}}{dt} = (BR6) + (BR9) + (BR16) - (BR12) - (BR20) - (BR28) - (BR34)
\end{cases}
\tag{54}
$$

It is assumed that the death rate of sensitive parasites in drug-treated hosts $\mu_{PW_D}$ is much larger than either $\mu_{PW_U}$ or $\mu_{PR_U}$, which are considered to be equal under the presumption that there is no fitness cost incurred on clearance rate in resistant parasites. In a situation where only one type of drug is being used in treatment, $\mu_{PR_D}$ is again equal or less than $\mu_{PW_U}$ and $\mu_{PR_U}$, but under a policy change or other use of a drug with different loci conferring resistance, $\mu_{PR_D}$ can be defined as the harmonic mean of two rates,

$$
\mu_{P_{R,D}} = 1/(Q/(\mu_{P_{R,U}}) + (1 - Q)/(\mu_{P_{W,D}}))
\tag{55}
$$

where $Q$ is the proportion of drug treatments using the drug to which resistance is being investigated.

### Immune memory submodel

The immune memory of the system is tracked as the total number of infections ($TI$) experienced by the hosts in $G_0, G - 1$, and $G_2$ classes via three ODEs. The general approach to writing the ODEs for these variables is to add the rate at which generalized immunity is gained, accounting for the movements of hosts between different $G$ classes. Furthermore, it is not necessary to track the immune memory to resistant and sensitive parasites separately because the resistance status does not influence the gaining of specific or generalized immunity. The rates used in the ODEs for the immune memory classes are the following:

**Increase of cleared infections**

$$
\begin{cases}
C1 = H_{0,U}(B_U(1 - \dfrac{r_{0,U}}{K})(1 - \eta_0) + r_{0,U}\mu_{PW_U}) + H_{0,D}(B_D(1 - \dfrac{r_{0,D}}{K})(1 - \eta_0) + r_{0,D}\mu_{PR_D}) \\[2mm]
C2 = H_{1,U}(B_U(1 - \dfrac{r_{1,U}}{K})(1 - \eta_1) + r_{1,U}\mu_{PW_U}) + H_{1,D}(B_D(1 - \dfrac{r_{1,D}}{K})(1 - \eta_1) + r_{1,D}\mu_{PR_D}) \\[2mm]
C3 = H_{2,U}(B_U(1 - \dfrac{r_{2,U}}{K})(1 - \eta_2) + r_{2,U}\mu_{PW_U}) + H_{2,D}(B_D(1 - \dfrac{r_{2,D}}{K})(1 - \eta_2) + r_{2,D}\mu_{PR_D})
\end{cases}
\tag{56}
$$

**Loss of infections counts due to immunity loss and host death**

$$\begin{cases} C4 = (\Lambda + \alpha)TI_0 \\ C5 = (\Lambda + \alpha)TI_1 \\ C6 = (\Lambda + \alpha)TI_2 \end{cases} \tag{57}$$

**Immunity carried by hosts moved from lower $G$ to higher $G$**

$$\begin{cases} C7 = (H_{0,U}(B_U(1 - \frac{r_{0,U}}{K})\rho_1(1 - \eta_0) + \rho_1 r_{0,U}\mu_{PW_U}) + H_{0,D}(B_D(1 - \frac{r_{0,D}}{K})\rho_1(1 - \eta_0) + \rho_1 r_{0,D}\mu_{PR_D}))\nu_0 \\ C8 = (H_{1,U}(B_U(1 - \frac{r_{1,U}}{K})\rho_2(1 - \eta_1) + \rho_2 r_{1,U}\mu_{PW_U}) + H_{1,D}(B_D(1 - \frac{r_{1,D}}{K})\rho_2(1 - \eta_1) + \rho_2 r_{1,D}\mu_{PR_D}))\nu_1 \end{cases}$$
$$\tag{58}$$

**Immunity carried by hosts moved from higher $G$ to lower $G$**

$$\begin{cases} C9 = (H_{1_U}(h_{1,U}\frac{\exp(-\frac{h_{1,U}}{\Lambda})}{1 - \exp(-\frac{h_{1,U}}{\Lambda})}) + H_{1_D}(h_{1,D}\frac{\exp(-\frac{h_{1,D}}{\Lambda})}{1 - \exp(-\frac{h_{1,D}}{\Lambda})}))\nu_1 \\ C10 = (H_{2,U}(h_{2,U}\frac{\exp(-\frac{h_{2,U}}{\Lambda})}{1 - \exp(-\frac{h_{2,U}}{\Lambda})}) + H_{2,D}(h_{2,D}\frac{\exp(-\frac{h_{2,D}}{\Lambda})}{1 - \exp(-\frac{h_{2,D}}{\Lambda})}))\nu_2 \end{cases} \tag{59}$$

The three ODEs describing the immune memory of the hosts are

$$\begin{cases} \frac{dTI_0}{dt} = (C1) + (C9) - (C4) - (C7) \\\\ \frac{dTI_1}{dt} = (C2) + (C7) + (C10) - (C5) - (C9) - (C8) \\\\ \frac{dTI_2}{dt} = (C3) + (C8) - (C6) - (C10) \end{cases} \tag{60}$$

Solving these ODEs gives the number of cumulative infections experienced by members of that host class at time $t$. The average number of such infections experienced by a member of the host class is $\nu_i$ at time $t$ such that

$$\begin{cases} \nu_0 = \frac{TI_0}{H_{0_U} + H_{0_D}} \\\\ \nu_1 = \frac{TI_1}{H_{1_U} + H_{1_D}} \\\\ \nu_2 = \frac{TI_2}{H_{2_U} + H_{2_D}} \end{cases} \tag{61}$$

These $\nu$ values can then be used in *Equation 1* to determine the infectivities of each host class (the $\eta_i$ values).

## Final implementation

A final addition to the model was to add the potential for seasonal variation in contact rate $b$. $b$ was set equal to a continuous, differentiable function of time:

$$b(t) = -b_{avg}Lcos(\kappa sin(\frac{t + p_s}{a}) - \frac{t + p_s}{a}) + b_{avg} \tag{62}$$

where $b_{avg}$ is the baseline daily transmission potential; $L$, on an interval of 0–1, is the amplitude of seasonal fluctuations relative to $b_{avg}$; $\kappa$ which is only biologically reasonable on the interval –1–1, controls the relative length of the dry season; $p_s$ is the phase shift; and $a$ is the period of oscillations. It should be noted that this approach to describing seasonal fluctuations is limited in how long of a dry season it can describe (because $\kappa > 1$ leads to a function with no biological implications).

A generalized immunity-only model was then specified as a counterpoint to the model incorporating specific immunity; in this model, the values of $\eta_i$ were decoupled from the accumulated infections and were set equal to a fixed value: $\eta_0 = 0.927, \eta_1 = 0.685, \eta_2 = 0.317$. These values were

the mean infectivities of the immunity classes over the range of parameter space at equilibrium from our full model. This procedure implies that there is no limit from strain-specific immune memory on infectivity.

The entire system of ODEs was solved numerically for the range of parameter values listed in *Appendix 1—table 1* using the package deSolve (*Soetaert et al., 2010*) in R (*R Development Core Team, 2023*), and the results were plotted and analyzed using packages tidyverse (*Wickham et al., 2019*) and ggplot2 (*Wickham, 2016*), also in R.

**Appendix 1—table 1.** Epidemiological parameters used for numerically solving ordinary differential equations (ODEs).

All rates are measured per day; time is measured in days.

| Symbol | Type | Description | Values |
|---|---|---|---|
| $b_{avg}$ | Rate | Baseline transmission potential | [0.0126–10] |
| $n_{strains}$ | Number | Local strain diversity | [6-447] |
| $\tau$ | Time | Period of drug effectiveness | 20 |
| $\omega$ | Probability | Chance of symptoms for $G_2$ | 0.01 |
| $\rho_1$ | Probability | Transition probability to $G_1$ | 0.5 |
| $\rho_2$ | Probability | Transition probability to $G_2$ | 0.05 |
| $s_{single}$ | Number | Clonal cost of resistance | [0–0.3] |
| $s_{mixed}$ | Number | Mixed-infection cost of resistance | [0–0.9] |
| $K$ | Number | Carrying capacity | 10 |
| $\mu_{PW_U}$ | Rate | $W_U$ clearance rate | 1/150 |
| $\mu_{PW_D}$ | Rate | $W_D$ clearance rate | 0.99 |
| $\mu_{PR_U}$ | Rate | $R_U$ clearance rate | 1/150 |
| $\mu_{PR_D}$ | Rate | $R_D$ clearance rate | 1/150 |
| $d_i$ | Rate | Daily treatment rate of symptomatic | [0.05,0.2] |
| $L$ | Scale factor | Amplitude of seasonal fluctuations | 0.95 |
| $\kappa$ | Scale factor | Relative length of dry season | 1 |
| $p_s$ | Number | Phase shift of seasonal fluctuations | 0 |
| $a$ | Time | Period of seasonal fluctuations | 365/(2π) |
| $Q$ | Proportion | Proportion of drug treatments using focus drug | [0–1] (see *Figure 7* caption) |
| $\delta$ | Rate | Daily constant host birth rate | 1 host |
| $\mu_{0,U}$ | Rate | Daily death rate of 0U hosts due to malaria | 1/1500 |
| $\mu_{0,D}$ | Rate | Daily death rate of 0D hosts due to malaria | 1/1500 |
| $\alpha$ | Rate | Death rate in the absence of malaria | 1/ (50*365) |
| $\Lambda$ | Rate | Rate of immunity loss | 0.001 |

