## [Editor Report · eLife assessment]

The study is an **important** advancement to the consideration of antimalarial drug resistance: the authors make use of both modeling results and supporting empirical evidence to demonstrate the role of malaria strain diversity in explaining biogeographic patterns of drug resistance. The theoretical methods and the corresponding results are **compelling**, with the novel model presented moving beyond existing models to incorporate malaria strain diversity and antigen-specific immunity. This work is likely to be interesting to malaria researchers and others working with antigenically diverse infectious diseases.

---

## [Referee Report · Reviewer #1 (Public Review)]

Summary:

The paper is an attempt to explain a geographic paradox between infection prevalence and antimalarial resistance emergence. The authors developed a compartmental model that importantly contains antigenic strain diversity and in turn antigen-specific immunity. They find a negative correlation between parasite prevalence and the frequency of resistance emergence and validate this result using empirical data of chloroquine-resistance. Overall, the authors conclude that strain diversity is a key player in explaining observed patterns of resistance evolution across different geographic regions.

The authors pose and address the following specific questions:

1. Does strain diversity modulate the equilibrium resistance frequency given different transmission intensities?

2. Does strain diversity modulate the equilibrium resistance frequency and its changes following drug withdrawal?

3. Does the model explain biogeographic patterns of drug resistance evolution?

Strengths:

The model built by the authors is novel. As emphasized in the manuscript, many factors (e.g., drug usage, vectorial capacity, population immunity) have been explored in models attempting to explain resistance emergence, but strain diversity (and strain specific immunity) has not been explicitly included and thus explored. This is an interesting oversight in previous models, given the vast antigenic diversity of *Plasmodium falciparum* (the most common human malaria parasite) and its potential to "drive key differences in epidemiological features".

The model also accounts for multiple infections, which is a key feature of malarial infections, with individuals often infected with either multiple Plasmodium species or multiple strains of the same species. Accounting for multiple infections is critical when considering resistance emergence, as with multiple infections there is within-host competition which will mediate the fitness of resistant genotypes. Overall, the model is an interesting combination of a classic epidemiological model (e.g., SIR) and a population genetics model.

In terms of major model innovations, the model also directly links selection pressure via drug administration with local transmission dynamics. This is accomplished by the interaction between strain-specific immunity, generalized immunity and host immune response.

Weaknesses:

The authors emphasize several model limitations, including the specification of resistance by a single locus (thus not addressing the importance of recombination should resistance be specified by more than one locus); the assumption that parasites are independently and randomly distributed among hosts (contrary to empirical evidence); and the assumption of a random association between the resistant genotype and antigenic diversity. However, each of these limitations are addressed in the discussion.

Did the authors achieve their goals? Did the results support their conclusion?

Returning to the questions posed by the authors:

1. Does strain diversity modulate the equilibrium resistance frequency given different transmission intensities? Yes. The authors demonstrate a negative relationship between prevalence/strain diversity and resistance frequency (Figure 2).

2. Does strain diversity modulate the equilibrium resistance frequency and its changes following drug withdrawal? Yes. The authors find that, under resistance invasion and some level of drug treatment, resistance frequency decreased with the number of strains (Figure 4). The authors also find that lower strain diversity results in a slower decline in resistant genotypes after drug withdrawal and higher equilibrium resistance frequency (Figure 6).

3. Does the model explain biogeographic patterns of drug resistance evolution? Yes. The authors find that their full model (which includes strain-specific immunity) produces the empirically observed negative relationship between resistance and prevalence/strain diversity, while a model only incorporating generalised immunity does not (Figure 8).

Utility of work to others and relevance within and beyond the field?

This work is important because antimalarial drug resistance has been an ongoing issue of concern for much of the 20th century and now 21st century. Further, this resistance emergence is not equitably distributed across biogeographic regions, with South America and Southeast Asia experiencing much of the burden of this resistance emergence. Not only can widespread resistant strains be traced back to these two relatively low-transmission regions, but these strains remain at high frequency even after drug treatment ceases.

---

## [Referee Report · Reviewer #2 (Public Review)]

Summary:

The evolution of resistance to antimalarial drugs follows a seemingly counterintuitive pattern, in which resistant strains typically originate in regions where malaria prevalence is relatively low. Previous investigations have suggested that frequent exposures in high-prevalence regions produce high levels of partial immunity in the host population, leading to subclinical infections that go untreated. These subclinical infections serve as refuges for sensitive strains, maintaining them in the population. Prior investigations have supported this hypothesis; however, many of them excluded important dynamics, and the results cannot be generalized. The authors have taken a novel approach using a deterministic model that includes both general and adaptive immunity. They find that high levels of population immunity produce refuges, maintaining the sensitive strains and allowing them to outcompete resistant strains. While general population immunity contributed, adaptive immunity is key to reproducing empirical patterns. These results are robust across a range of fitness costs, treatment rates, and resistance efficacies. Given sufficient antigenic diversity and high transmission, sensitive parasites remain in circulation even when there is no cost to resistance. This work demonstrates that future investigations cannot overlook adaptive immunity and antigenic diversity.

Strengths:

Overall, this is a very nice paper that makes a significant contribution to the field. It is well-framed within the body of literature and achieves its goal of providing a generalizable, unifying explanation for otherwise disparate investigations. The model is innovative. The approach is elegant and rigorous, with results that are supported across a broad range of parameters when considered within an equilibrium setting. Their exploration of geographical patterns of resistance makes the results of their simulations even more compelling. As such, this work will likely serve as a foundation for many future investigations.

Weaknesses:

Although the authors model resistance invasion, it does not align with empirical observations of the spread of resistance. For example, Plasmodium's mutation rate and population size mean that mutations providing chloroquine resistance should arise repeatedly even within a single infection. Nevertheless, Africa remained free of chloroquine resistant strains until a lineage was introduced from Asia. Upon introduction, it spread across the continent within ten years. The difference between the fate of chloroquine resistance originating in Africa versus chloroquine resistance originating in Asia cannot be attributed to changes in population immunity and treatment.

The source of this disparity may be in part attributable to the use of a deterministic, compartmental model, as the authors mention in the discussion. Strains are not explicitly modeled. This means that in terms of the distribution of strain diversity, the resistant and the sensitive compartments are identical, and the locus determining resistance is equally distributed across all strain backgrounds. However, substantial rates of linkage disequilibrium and clonal reproduction are found even in high transmission settings. The model assumptions may be met at equilibrium, but are not appropriate for most scenarios involving the invasion of a rare mutation.

---

## [Author Response]

The following is the authors’ response to the original reviews.

**eLife assessment**
The study is an important advancement to the consideration of antimalarial drug resistance: the authors make use of both modelling results and supporting empirical evidence to demonstrate the role of malaria strain diversity in explaining biogeographic patterns of drug resistance. The theoretical methods and the corresponding results are convincing, with the novel model presented moving beyond existing models to incorporate malaria strain diversity and antigen-specific immunity. This work is likely to be interesting to malaria researchers and others working with antigenically diverse infectious diseases.
**Public Reviews:**

**Reviewer #1 (Public Review):**
Summary:The paper is an attempt to explain a geographic paradox between infection prevalence and antimalarial resistance emergence. The authors developed a compartmental model that importantly contains antigenic strain diversity and in turn antigen-specific immunity. They find a negative correlation between parasite prevalence and the frequency of resistance emergence and validate this result using empirical data on chloroquine-resistance. Overall, the authors conclude that strain diversity is a key player in explaining observed patterns of resistance evolution across different geographic regions.The authors pose and address the following specific questions:1. Does strain diversity modulate the equilibrium resistance frequency given different transmission intensities?1. Does strain diversity modulate the equilibrium resistance frequency and its changes following drug withdrawal?1. Does the model explain biogeographic patterns of drug resistance evolution?Strengths:The model built by the authors is novel. As emphasized in the manuscript, many factors (e.g., drug usage, vectorial capacity, population immunity) have been explored in models attempting to explain resistance emergence, but strain diversity (and strain-specific immunity) has not been explicitly included and thus explored. This is an interesting oversight in previous models, given the vast antigenic diversity of *Plasmodium falciparum* (the most common human malaria parasite) and its potential to "drive key differences in epidemiological features".The model also accounts for multiple infections, which is a key feature of malarial infections, with individuals often infected with either multiple Plasmodium species or multiple strains of the same species. Accounting for multiple infections is critical when considering resistance emergence, as with multiple infections there is within-host competition which will mediate the fitness of resistant genotypes. Overall, the model is an interesting combination of a classic epidemiological model (e.g., SIR) and a population genetics model.In terms of major model innovations, the model also directly links selection pressure via drug administration with local transmission dynamics. This is accomplished by the interaction between strain-specific immunity, generalized immunity, and host immune response.

R: We thank the reviewer for his/her appreciation of the work.

Weaknesses:In several places, the explanation of the results (i.e., why are we seeing this result?) is underdeveloped. For example, under the section "Response to drug policy change", it is stated that (according to the model) low diversity scenarios show the least decline in resistant genotype frequency after drug withdrawal; however, this result emerges mechanistically. Without an explicit connection to the workings of the model, it can be difficult to gauge whether the result(s) seen are specific to the model itself or likely to be more generalizable.

R: We acknowledge that the explanation of certain results needs to be improved. We have now added the explanation of why low diversity scenarios show the least decline in resistance frequency after drug withdrawal: “Two processes are responsible for the observed trend: first, resistant genotypes have a much higher fitness advantage in low diversity regions even with reduced drug usage because infected hosts are still highly symptomatic; second, due to low transmission potential in low diversity scenarios (i.e., longer generation intervals between transmissions), the rate of change in parasite populations is slower.” (L243-247). We also compared the drug withdrawal response to that of the generalized-immunity-only model (L268-271). The medium transmission region has the fastest reduction in resistance frequency, followed by the high and low transmission regions, which differs from the full model that incorporates strain-specific diversity.

In addition, to provide the context of different biogeographic transmission zones, we now include a new figure (now Fig. 3) that presents the parameter space of transmission potential and strain diversity of different continents, which demonstrates that PNG and South America have less strain diversity than expected by transmission potential (L179-184 and L198-202). Therefore, these two regions have low disease prevalence and high resistance frequency.

The authors emphasize several model limitations, including the specification of resistance by a single locus (thus not addressing the importance of recombination should resistance be specified by more than one locus); the assumption that parasites are independently and randomly distributed among hosts (contrary to empirical evidence); and the assumption of a random association between the resistant genotype and antigenic diversity. However, each of these limitations is addressed in the discussion.

R: As pointed out by the referee, our model presents several limitations that have all been addressed in the discussion and considered for future extensions.

Did the authors achieve their goals? Did the results support their conclusion?Returning to the questions posed by the authors:1. Does strain diversity modulate the equilibrium resistance frequency given different transmission intensities? Yes. The authors demonstrate a negative relationship between prevalence/strain diversity and resistance frequency (Figure 2).1. Does strain diversity modulate the equilibrium resistance frequency and its changes following drug withdrawal? Yes. The authors find that, under resistance invasion and some level of drug treatment, resistance frequency decreased with the number of strains (Figure 4). The authors also find that lower strain diversity results in a slower decline in resistant genotypes after drug withdrawal and higher equilibrium resistance frequency (Figure 6).1. Does the model explain biogeographic patterns of drug resistance evolution? Yes. The authors find that their full model (which includes strain-specific immunity) produces the empirically observed negative relationship between resistance and prevalence/strain diversity, while a model only incorporating generalised immunity does not (Figure 8).Utility of work to others and relevance within and beyond the field?This work is important because antimalarial drug resistance has been an ongoing issue of concern for much of the 20th century and now 21st century. Further, this resistance emergence is not equitably distributed across biogeographic regions, with South America and Southeast Asia experiencing much of the burden of this resistance emergence. Not only can widespread resistant strains be traced back to these two relatively low-transmission regions, but these strains remain at high frequency even after drug treatment ceases.
**Reviewer #2 (Public Review):**
Summary:The evolution of resistance to antimalarial drugs follows a seemingly counterintuitive pattern, in which resistant strains typically originate in regions where malaria prevalence is relatively low. Previous investigations have suggested that frequent exposures in high-prevalence regions produce high levels of partial immunity in the host population, leading to subclinical infections that go untreated. These subclinical infections serve as refuges for sensitive strains, maintaining them in the population. Prior investigations have supported this hypothesis; however, many of them excluded important dynamics, and the results cannot be generalized. The authors have taken a novel approach using a deterministic model that includes both general and adaptive immunity. They find that high levels of population immunity produce refuges, maintaining the sensitive strains and allowing them to outcompete resistant strains. While general population immunity contributed, adaptive immunity is key to reproducing empirical patterns. These results are robust across a range of fitness costs, treatment rates, and resistance efficacies. They demonstrate that future investigations cannot overlook adaptive immunity and antigenic diversity.

R: We thank the reviewer for his/her appreciation of the work.

Strengths:Overall, this is a very nice paper that makes a significant contribution to the field. It is well-framed within the body of literature and achieves its goal of providing a generalizable, unifying explanation for otherwise disparate investigations. As such, this work will likely serve as a foundation for future investigations. The approach is elegant and rigorous, with results that are supported across a broad range of parameters.Weaknesses:Although the title states that the authors describe resistance invasion, they do not support or even explore this claim. As they state in the discussion (line 351), this work predicts the equilibrium state and doesn't address temporal patterns. While refuges in partially immune hosts may maintain resistance in a population, they do not account for the patterns of resistance spread, such as the rapid spread of chloroquine resistance in Africa once it was introduced from Asia.

R: We do agree that resistance invasion is not the focus of our manuscript. Rather we mainly investigate the maintenance and decline after drug withdrawal. Therefore, we changed the title to “Antigenic strain diversity predicts different biogeographic patterns of maintenance and decline of anti-malarial drug resistance” (L1-4).

We did, however, present a fast initial invasion phase for the introduction of resistant genotypes regardless of transmission scenarios in Fig. 5 (now Fig. 6). Even though the focus of the manuscript is to investigate long term persistence of resistant genotypes, we did emphasize that the initial invasion phase and how that changes the host immunity profile are key to the coexistence of resistant and wild-type genotypes (L228-239).

As the authors state in the discussion, the evolution of compensatory mutations that negate the cost of resistance is possible, and in vitro experiments have found evidence of such. It appears that their results are dependent on there being a cost, but the lower range of the cost parameter space was not explored.

R: It is true that compensatory mutations might mitigate the negative fitness consequences. We didn’t add a no-cost scenario because in general if there is no cost but only benefit (survival through drug usage), then resistant haplotypes will likely be fixed in the population. This is contingent on the assumption that these compensatory mutations are in perfect linkage with resistant alleles, which is unlikely in high-transmission scenarios. Our model does not incorporate recombination, but earlier models (Dye & Williams 1997, Hastings & D’Alessandro 2000) have demonstrated that recombination will delay the fixation of resistant alleles in high-transmission.

As suggested, we ran our model with costs equal 0 and 0.01 (Fig. 2C and L189-191). We found that resistant alleles almost always fix except for when diversity is extremely high, treatment/resistance efficacy is low. In these cases, additional benefits brought by more transmission from resistant alleles do not bring many benefits (as lower GI classes have a very small number of hosts). This finding does not contradict a wider range of coexistence between wild-type and resistant alleles when the cost is higher. We therefore added these scenarios to our updated results.

The use of a deterministic, compartmental model may be a structural weakness. This means that selection alone guides the fixation of new mutations on a semi-homogenous adaptive landscape. In reality, there are two severe bottlenecks in the transmission cycle of Plasmodium spp., introducing a substantial force of stochasticity via genetic drift. The well-mixed nature of this type of model is also likely to have affected the results. In reality, within-host selection is highly heterogeneous, strains are not found with equal frequency either in the population or within hosts, and there will be some linkage between the strain and a resistance mutation, at least at first. Of course, there is no recourse for that at this stage, but it is something that should be considered in future investigations.

R: We thank the reviewer for their insightful comments on the constraints of the deterministic modeling approach. We’ve added these points to discussion in the paragraph discussing the second limitation of the model (L359-364).

The authors mention the observation that patterns of resistance in high-prevalence Papua New Guinea seem to be more similar to Southeast Asia, perhaps because of the low strain diversity in Papua New Guinea. However, they do not investigate that parameter space here. If they did and were able to replicate that observation, not only would that strengthen this work, it could profoundly shape research to come.

R: We appreciate the suggestion to investigate the parameter space of Papua New Guinea. We now include a new figure (now Fig. 3) that presents the parameter space of transmission potential and strain diversity of different continents, which demonstrates that PNG and South America have less strain diversity than expected by transmission potential (L179-184 and L198-202). This translates to low infectivity for most mosquito bites, and most infections only occur in hosts with lower generalized immunity. Therefore resistant genotypes will help ensure disease transmission in these symptomatic hosts and be strongly selected to be maintained.

**Reviewer #1 (Recommendations For The Authors):**
1. I found lines 41-49 difficult to follow. Please rephrase (particularly punctuation) for clarity.

R: We have edited the lines to improve the writing (L41-50):

“Various relationships between transmission intensity and stable frequencies of resistance were discovered, each of which has some empirical support: (1) transmission intensity does not influence the fate of resistant genotypes [Models: Koella and Antia (2003); Masserey et al. (2022); Empirical: Diallo et al. (2007); Shah et al. (2011, 2015)]; (2) resistance first increases in frequency and slowly decreases with increasing transmission rates [Models: Klein et al. (2008, 2012)]; and (3) Valley phenomenon: resistance can be fixed at both high and low end of transmission intensity [Model: Artzy-Randrup et al. (2010); Empirical: Talisuna et al. (2002)]. Other stochastic models predict that it is harder for resistance to spread in high transmission regions, but patterns are not systematically inspected across the parameter ranges [Model: Whitlock et al. (2021); Model and examples in Ariey and Robert (2003)].”

1. Line 65: There should be a space after "recombination" and before the citation.

R: Thank you for catching the error. We’ve added the space (L64).

1. I'm interested in the dependency of the results on the assumption that there is a cost to resistance via lowered transmissibility (lines 142-145). I appreciate that variation in the cost(s) of resistance in single and mixed infections is explored; however, from what I can tell the case of zero cost is not explored.

R: As suggested, we have now added the no-cost scenario. Please see the response to the Reviewer2 weaknesses paragraph 2.

1. I felt the commentary/explanation of the response to drug policy change was a bit underdeveloped. I would have liked a walk-through of why in your model low diversity scenarios show the slowest decline in resistant genotypes after switching to different drugs.

R: We acknowledge that the explanation of the response to drug policy change needs to be improved. We have now added the explanation of why we observe low diversity scenarios show the least decline in resistance frequency after drug withdrawal: “Two processes are responsible for the seen trend: first, resistant genotypes have a much higher fitness advantage in low diversity regions even with reduced drug usage because infected hosts are still highly symptomatic; second, due to low transmission potential in low diversity scenarios (i.e., longer generation intervals between transmissions), the rate of change in parasite populations is slower.” (L243-247). We also compared the drug withdrawal response to that of the generalized-immunity-only model. The medium transmission region has the fastest reduction in resistance frequency, followed by the high and low transmission regions, which differs from the full model that incorporates strain-specific diversity.

1. Line 352: persistent drug usage?

R: Yes, we meant persistent drug usage. We’ve clarified the writing (L389-391).

1. The organisation of the manuscript would benefit from structuring around the focal questions so that the reader can easily find the answers to the focal questions within the results and discussion sections.

R: This is a great suggestion. We modified the subheadings of results to provide answers to focal questions (L151, L179, L203-204, and L240).

1. Line 353: Please remove either "shown" or "demonstrated".

R: Thank you for catching the grammatical error, we’ve retained “shown” only for the sentence (L391-392).

**Reviewer #2 (Recommendations For The Authors):**
Overall, this was very nice work and a pleasure to read.Major:1. Please provide a much more thorough explanation of how resistance invasions are modeled. It is not clear from the text and could not be replicated.

R: We have now added a section “drug treatment and resistance invasion” in Methods and Materials to explain how resistance invasions are modeled (L488-496):

“Given each parameter set, we ran the ODE model six times until equilibrium with the following genotypic compositions: (1) wild-type only scenario with no drug treatment; (2) wild-type only scenario with 63.2% drug treatment (0.05 daily treatment rate); (3) wild-type only scenario with 98.2% drug treatment (0.2 daily treatment rate); (4) resistant-only scenario with no drug treatment; (5) resistance invasion with 63.2% drug treatment; (6) resistance invasion with 98.2% drug treatment. Runs 1-4 start with all hosts in G0,U compartment and ten parasites. Runs 5 and 6 (resistance invasion) start from the equilibrium state of 2 and 3, with ten resistant parasites introduced. We then followed the ODE dynamics till the next equilibrium.”

1. Please make your raw data, code, and replicable examples that produce the figures in the manuscript available.

R: We have added the data availability session, which provides the GitHub site with all the code for the model, data processing, and figures: All the ODE codes, numerically-simulated data, empirical data, and analyzing scripts are publicly available at https://github.itap.purdue.edu/HeLab/MalariaResistance.

1. Regarding the limitations described in the paragraph about the model in the public response, these results would be strengthened if there were separate compartments for strains which could be further divided into sensitive and resistant. Could you explore this for at least a subset of the parameter space?

R: In our model, sensitive and resistant pathogens are always modeled as separate compartments (Fig. S1B and Appendix 1). In Results/Model structure, L135-136, we stated the setup:

“The population sizes of resistant (PR) or sensitive (wild-type; PW) parasites are tracked separately in host compartments of different G and drug status.”

1. To what extent do these results rely on a cost to resistance? Were lower costs explored? This would be worth demonstrating. If this cannot be maintained without cost, do you think this is because there is no linkage between strain and resistance?

R: As suggested, we have now added the no-cost scenario (Fig. 2C and L189-191). Please see the response to the Reviewer1 weaknesses paragraph 2. In sum, under a no-cost scenario, if treatment rate is low, then wild-type alleles will still be maintained in high transmission scenarios; when treatment rate is high, resistant alleles will always be fixed.

Minor:1. "Plasmodium" should be italicized throughout. Ironically, italics aren't permitted in this form.

R: We did italicize “Plasmodium” or “*P. falciparum*” throughout the text. If the reviewer is referring to “falciparum malaria”, the convention is not to italicize falciparum in this case.

1. Fig 1A: the image is reversed for the non-infected host with prior exposure to strain A. Additionally, the difference between colors for WT and resistant is not visible in monochrome.

R: Thank you for pointing out the problem of color choice in monochrome. We have modified the figure. The image in Fig 1A is not reversed for non-infected hosts with prior exposure to strain A. We now spell out “S” to be “specific immunity”, and explain it better in the figure legend.

1. Fig 2B: add "compare to the pattern of prevalence shown in Fig 2A" or something similar to make the comparison immediately clear.

R: We thank the reviewer’s suggestion. We’ve added a sentence to contrast Fig 2A and B in the Figure legend: “A comparison between the prevalence pattern in (A) and resistance frequency in (B) reveals that high prevalence regions usually correspond to low resistance frequency at the end of resistance invasion dynamics.”

1. Figs 2B & C: Please thoroughly explain how you produced this data in the methods section and briefly describe it in the results sections.

R: We agree that the modeling strategies need to be explained better. Since we explained the rationale for the parameter ranges and the prevalence patterns we observe in the results section “Appropriate pairing of strain diversity and vectorial capacity” (now “Impact of strain diversity and transmission potential on disease prevalence”), we added sentences in this section to explain how we run models until equilibrium for wild-only infections with or without drug treatment (L152-178). Then in the following section “Drug-resistance and disease prevalence” section, we explain how we obtained the resistance invasion data:

“To investigate resistance invasion, we introduce ten resistant infections to the equilibrium states of drug treatment with wild-type only infections, and follow the ODE dynamics till the next equilibrium” (L180-181).

1. Fig 3: The axis labels are not particularly clear. For the Y axis, please state in the label what it is the frequency of (either the mutation or the phenotype). In the X axis, it is better to spell that out in words, like "*P. falciparum* prevalence in children".

R: Thank you for pointing this out. We’ve modified the axes labels of Fig. 3 (now Fig. 4): X-axis: “*P. falciparum* prevalence in children aged 2-10”; Y-axis: “Frequency of resistant genotypes (pfcrt 76T)”.

1. Fig 4 and the rest of the figures of this nature: Showing an equilibrium-state timestep before treatment was introduced would improve the readers' understanding of the dynamics.

R: We agree that the equilibrium state before treatment is important. In fact, we have those states in our figure 4 (now figure 5): the left panel- “Daily treatment rate 0” indicates the equilibrium-state timestep before treatment. We clarified this point in the caption.

1. Fig 5 is very compelling, but the relationships in Fig 5 would be clearer if the Y axes were not all different. Consider using the same scale for the hosts, and the same scale for resistant parasites (both conditions) and WT parasites, 113 strains. It may be clearer to reference them if they are given as A-F instead of three figures each for A and B.

R: We agree with the suggested changes and have modified figure 5 (now Fig. 6): we used one Y-axis scale for the hosts, and one Y-axis scale for the parasites. The wild-type one is very low for the low diversity scenario, thus we included one inset plot for that case.

1. Fig 5 caption: High immune protection doesn't select against resistance. The higher relative fitness of the sensitive strain selects against resistance in a high-immunity environment.

R: Thank you for pointing this out. Here we meant that a reduction in resistant population after the initial overshoot occurs in both diversity levels. We are not comparing resistant strains to sensitive ones. We’ve modified the sentence to: “The higher specific immunity reduces the infectivity of new strains, leading to a reduction of the resistant parasite population regardless of the diversity level”.

1. Line 242: "keep" should be plural.

R: We’ve corrected “keep” to “keeps” (L267).

1. Line 360 and elsewhere: The strength of the results is somewhat overstated at times. This absolutely supports the importance of strain-specific immunity, but these results do not explain patterns of the origin of resistance and there are a number of factors that are not incorporated (a necessary evil of modeling to be sure).

R: Thank you for pointing this out. We’ve modified discussion to remove the overstated strength of results:

1. Original: “The inclusion of strain diversity in the model provides a new mechanistic explanation as to why Southeast Asia has been the original source of resistance to certain antimalarial drugs, including chloroquine.”

Modified: “The inclusion of strain diversity in the model provides a new mechanistic explanation as to why Southeast Asia has persisting resistance to certain antimalarial drugs, including chloroquine, despite a lower transmission intensity than Africa. “ (L328-330)

1. In sum, we show that strain diversity and associated strain-specific host immunity, dynamically tracked through the macroparasitic structure, can explainpredict the complex relationship between transmission intensity and drug-resistance frequencies.

1. The color palettes are not discernible in grayscale, especially the orange/blue/gray in Fig 2. The heatmaps appear to be in turbo, the only viridis palette that isn't grayscale-friendly. Just something to keep in mind for the accessibility of individuals with achromatopsia and most people who print out papers.

R: Thank you for the visualization suggestions. We updated all the figures with the “viridis:magma” palette. As for the orange/blue/gray scale used in Fig 2C, it is difficult to pick nine colors that are discernable in brightness in grayscale. Currently, the four colors correspond to clonal genotype cost (i.e. green, red, grey, and blue), and the three-level brightness maps to mixed genotype cost.